# The Molecular Heterogeneity of Store-Operated Ca^2+^ Entry in Vascular Endothelial Cells: The Different roles of Orai1 and TRPC1/TRPC4 Channels in the Transition from Ca^2+^-Selective to Non-Selective Cation Currents

**DOI:** 10.3390/ijms24043259

**Published:** 2023-02-07

**Authors:** Francesco Moccia, Valentina Brunetti, Angelica Perna, Germano Guerra, Teresa Soda, Roberto Berra-Romani

**Affiliations:** 1Laboratory of General Physiology, Department of Biology and Biotechnology “L. Spallanzani”, University of Pavia, 27100 Pavia, Italy; 2Department of Medicine and Health Sciences, University of Molise, 86100 Campobasso, Italy; 3Department of Health Sciences, University of Magna Graecia, 88100 Catanzaro, Italy; 4Department of Biomedicine, School of Medicine, Benemérita Universidad Autónoma de Puebla, Puebla 72410, Mexico

**Keywords:** vascular endothelial cells, store-operated Ca^2+^ entry, STIM, Orai, TRPC1, TRPC4, inositol-1,4,5-trisphosphate receptors, I_CRAC_, I_SOC_, I_CRAC_-like

## Abstract

Store-operated Ca^2+^ entry (SOCE) is activated in response to the inositol-1,4,5-trisphosphate (InsP_3_)-dependent depletion of the endoplasmic reticulum (ER) Ca^2+^ store and represents a ubiquitous mode of Ca^2+^ influx. In vascular endothelial cells, SOCE regulates a plethora of functions that maintain cardiovascular homeostasis, such as angiogenesis, vascular tone, vascular permeability, platelet aggregation, and monocyte adhesion. The molecular mechanisms responsible for SOCE activation in vascular endothelial cells have engendered a long-lasting controversy. Traditionally, it has been assumed that the endothelial SOCE is mediated by two distinct ion channel signalplexes, i.e., STIM1/Orai1 and STIM1/Transient Receptor Potential Canonical 1(TRPC1)/TRPC4. However, recent evidence has shown that Orai1 can assemble with TRPC1 and TRPC4 to form a non-selective cation channel with intermediate electrophysiological features. Herein, we aim at bringing order to the distinct mechanisms that mediate endothelial SOCE in the vascular tree from multiple species (e.g., human, mouse, rat, and bovine). We propose that three distinct currents can mediate SOCE in vascular endothelial cells: (1) the Ca^2+^-selective Ca^2+^-release activated Ca^2+^ current (I_CRAC_), which is mediated by STIM1 and Orai1; (2) the store-operated non-selective current (I_SOC_), which is mediated by STIM1, TRPC1, and TRPC4; and (3) the moderately Ca^2+^-selective, I_CRAC_-like current, which is mediated by STIM1, TRPC1, TRPC4, and Orai1.

## 1. Introduction

The endothelial monolayer forms the innermost lining of all blood vessels and of the cardiac chambers, where it is known as endocardium. Endothelial cells provide a widespread interface that continuously perceives and integrates signals delivered by blood-borne humoral cues and by changes in shear stress and/or pulsatile stretch. Additionally, endothelial cells may sense even subtle alterations in local tissue microenvironment and/or in the signalling activity of neighbouring cells [1,2,3,4,5]. Endothelial dysfunction is, therefore, involved in the pathogenesis of life-threatening cardiovascular disorders, such as coronary artery disease, acute myocardial infarction, atherosclerosis, hypertension, diabetes mellitus, stroke, and, more, recently COVID-19 [3,6,7]. Endothelial colony forming cells (ECFCs) can be mobilized from vascular stem cell niches to replace senescent/injured endothelial cells or reconstruct the vascular network damaged by an ischemic insult, thereby attempting to preserve cardiovascular homeostasis [8]. Insufficient or excessive vascularization by capillary endothelial cells and circulating ECFCs has been recognized as a common denominator of multiple debilitating or deadly diseases, including age-related blindness, acute myocardial infarction, and cancer [9,10].

A spatio-temporal increase in intracellular Ca^2+^ concentration ([Ca^2+^]_i_) is crucial to select the most appropriate endothelial response to the multiple inputs that are simultaneously delivered either from blood flow or from adjoining tissue [2,11,12,13]. Endothelial Ca^2+^ signals can adopt distinct waveforms, e.g., biphasic Ca^2+^ elevation or intracellular Ca^2+^ oscillations, to regulate a plethora of functions [2,14], such as vascular tone and blood pressure [15,16], vascular permeability [17,18], and repair [19,20], neurovascular coupling [4,21], platelet aggregation and blood coagulation [22], leukocyte infiltration [23], and angiogenesis [24]. Furthermore, an increase in [Ca^2+^]_i_ drives ECFCs’ angiogenic activity both in vitro [7,25] and in vivo [26,27]. Typically, the endothelial Ca^2+^ response to humoral stimulation is triggered by Ca^2+^ release from the endoplasmic reticulum (ER), i.e., the largest endogenous Ca^2+^ reservoir, through inositol-1,4,5-trisphosphate (InsP_3_) receptors (InsP_3_Rs) [24]. This may cause a dramatic drop in ER Ca^2+^ concentration ([Ca^2+^]_ER_), which results in the activation of a Ca^2+^-permeable pathway in the plasma membrane (PM), known as store-operated Ca^2+^ entry (SOCE) [28,29,30]. SOCE fulfils the primary role to refill the ER with Ca^2+^ and thereby maintains the intracellular Ca^2+^ response over time. Additionally, SOCE recruits the variety of Ca^2+^-dependent decoders that effect the endothelial response to extracellular stimulation [28,29,30]. Therefore, SOCE is regarded as the most important Ca^2+^ entry pathway in both vascular endothelial cells [28,29,30] and in circulating ECFCs [31].

Traditionally, it is assumed that depletion of the endothelial ER Ca^2+^ store may lead to the activation of two distinct inward currents that exhibit distinct electrophysiological features and are mediated by different cation channels on the PM. Store depletion in vascular endothelial cells could activate a Ca^2+^-selective and inwardly rectifying current that resembles the Ca^2+^ release-activated Ca^2+^ (CRAC) current (I_CRAC_) originally described in mast cells [32] and is, therefore, referred to as I_CRAC_ [33,34,35]. Alternately, the reduction in [Ca^2+^]_ER_ could result in the development of a non-selective cation current that displays a slightly linear current-to-voltage (IV) relationship and is known as store-operated current (I_SOC_) [36,37]. A flurry of investigations published over the last twenty years paved the way towards the general agreement that, in vascular endothelial cells, Stromal Interaction Molecule 1 (STIM1) and Orai1 proteins mediate the I_CRAC_ [38,39,40], whereas members of the Transient Receptor Potential (TRP) Canonical (TRPC) subfamily of non-selective cation channels underlie the I_SOC_ [41,42,43]. A careful scrutiny of the literature, however, reveals that depletion of the ER Ca^2+^ store can activate a third endothelial Ca^2+^-entry pathway, which is more Ca^2+^ selective than the I_SOC_ but displays a weaker inward rectification and has a more negative reversal potential (E_rev_) as respect to the I_CRAC_ [44,45,46,47]. This current has been termed as I_SOC_ [48] or I_CRAC_-like [46,49]. In our opinion, the latter definition more closely depicts the electrophysiological properties of this store-dependent current, whose molecular architecture has been recently unveiled and consists of both Orai and TRP channel proteins [50,51]. Therefore, herein, we aim at bringing order to the distinct mechanisms whereby endothelial SOCE can occur and modulate different functions in the vascular tree from multiple species (e.g., human, mouse, rat, and bovine).

## 2. How to Activate Endothelial SOCE: Physiological vs. Pharmacological Reduction in [Ca^2+^]_ER_

Endothelial SOCE can be activated either by physiological cues that stimulate membrane receptors to liberate ER stored Ca^2+^ via InsP_3_Rs or by pharmacological compounds that mobilize intraluminal Ca^2+^ and thereby increase the membrane permeability to Ca^2+^ through receptor-independent mechanisms [49,52].

### 2.1. The Physiological Activation of SOCE by InsP_3_-Dependent ER Ca^2+^ Release

Physiologically, the endothelial Ca^2+^ store is depleted upon extracellular stimulation by humoral agonists, such as hormones, neurotransmitters, and growth factors, which bind to their specific G_q_/11 protein-coupled receptors (G_q_PCRs) or tyrosine kinase receptors (TKRs). G_q_PCRs and TKRs, respectively, engage phospholipase Cβ (PLCβ) and PLCγ that cleave phosphatidylinositol 4,5-bisphosphate (PIP_2_), a minor phospholipid residing in the inner leaflet of the PM, in diacylglycerol (DAG) and InsP_3_ [24,53]. InsP_3_ diffuses in the cytoplasm and primes ER-embedded InsP_3_Rs towards ambient Ca^2+^, thereby mobilizing ER Ca^2+^ through the mechanism of Ca^2+^-induced Ca^2+^ release (CICR) [24,53]. The local Ca^2+^ pulse that is required by InsP_3_ to engage the Ca^2+^-dependent InsP_3_Rs can be provided by the closely apposed two-pore channels (TPCs), that release endolysosomal Ca^2+^ in response to nicotinic acid adenine dinucleotide phosphate (NAADP) [54,55]. Conversely, ryanodine receptors (RyRs) play a minor role in ER Ca^2+^ release in endothelial cells, as recently discussed in [14].

The mechanisms whereby the stimulation of G_q_PCRs leads to the InsP_3_-dependent depletion of the ER Ca^2+^ store have been finely dissected in calf pulmonary artery endothelial (CPAE) cells [28]. Agonist-induced elevation in [Ca^2+^]_i_ arises in the form of spatially restricted InsP_3_-dependent ER Ca^2+^ release events, known as Ca^2+^ puffs, that arise in the cell periphery and subsequently propagate as repetitive Ca^2+^ waves to engulf the whole cytoplasm through CICR [56,57]. Notably, focal stimulation with an InsP_3_-producing agonist (i.e., ATP) may result in local depletion of the InsP_3_-sensitive ER Ca^2+^ store, which leads to SOCE activation both at the stimulated site and, following repetitive stimulations, at more distant regions where no Ca^2+^ release events has occurred [58]. An increase in ATP concentration caused a dose-dependent reduction in [Ca^2+^]_ER_ that induced a gradual increase in SOCE activity; furthermore, SOCE underwent periodic cycles of rapid activation and slow deactivation during the slow intracellular Ca^2+^ oscillations that can be elicited by prolonged stimulation of CPAE cells with a submaximal concentration of ATP [59]. These observations were recently confirmed in human umbilical vein endothelial cells (HUVECs), where local PLC activity generates spatially-restricted Ca^2+^ pulses at the front edge of migrating cells, thereby causing local ER Ca^2+^ depletion and SOCE activation [60].

### 2.2. The Pharmacological Activation of SOCE by ER Ca^2+^ Mobilization

SOCE can also be activated by the pharmacological depletion of the ER Ca^2+^ by means of multiple drugs. The most common strategy to evoke SOCE in endothelial cells (and in circulating ECFCs) is represented by the blockade of Sarco-Endoplasmic Reticulum Ca^2+^ (SERCA) activity with cyclopiazonic acid (CPA) [61,62,63,64], a mycotoxin and a fungal neurotoxin produced by the moulds Aspergillus and Penicillium and by thapsigargin [38,39,51,63], a plant-derived lactone, whereas 2,5-di-tert-butylhydroquinone (tBHQ) was mostly exploited in earlier studies [65,66,67]. The blockade of SERCA activity results in passive ER Ca^2+^ efflux from leakage channels, whose molecular identity in endothelial cells remains to be unveiled [68]. The following reduction in [Ca^2+^]_ER_ is sufficient to activate SOCE on the PM in an InsP_3_-independent manner [52]. Extracellular application of thapsigargin (whose inhibitory effect on SERCA is irreversible) has routinely been exploited to detect SOCE by whole-cell patch-clamp electrophysiology [38,39,43,46,69], while both CPA and thapsigargin may be used to elicit SOCE during Ca^2+^ imaging recordings [38,39,61,62,63,64]. An alternative strategy to induce store-operated inward currents is through the intracellular infusion of InsP_3_ (to induce ER Ca^2+^ depletion) and/or a high concentration (e.g., 20 mM) of BAPTA (to buffer Ca^2+^ leaking out from the ER) through the patch pipette [35,38,43]. The Ca^2+^ ionophore, ionomycin, can also activate endothelial SOCE [70,71,72], but its use has been limited by the evidence that this drug can increase the Ca^2+^ permeability of other organelles, such as mitochondria [73].

### 2.3. The Search for the Coupling Mechanism between ER Ca^2+^ Depletion and SOCE Activation in Vascular Endothelial Cells

The mechanisms whereby a reduction in [Ca^2+^]_ER_ leads to SOCE activation in endothelial cells [28,30], as well as in both non-excitable [52] and excitable cells [74], have been object of intense scrutiny over the years. A groundbreaking discovery was published in 2005, when Roos et al. demonstrated that stromal interaction molecule 1 (STIM1) was crucial to elicit SOCE [75]. Shortly thereafter, Liou et al. showed that STIM1 served as the sensor of ER Ca^2+^ concentration ([Ca^2+^]_ER_) that is activated upon ER Ca^2+^ emptying [76]. Subsequent investigations rapidly demonstrated that STIM1 could interact with the newly discovered Orai channels to mediate the classical I_CRAC_ [77,78,79] or with members of the TRPC sub-family of non-selective cation channels, e.g., TRPC1, to mediate the I_SOC_ [80,81,82]. Interestingly, STIM1, Orai1, and TRPC1 can interact either directly, i.e., by forming supermolecular ternary complexes [80,83,84], or functionally, e.g., through Orai1-dependent insertion of TRPC1 channels on the PM [85]. The potential interaction of the STIM1-gated Ora1 and TRPC1 channels, which display distinct Ca^2+^ permeabilities and E_rev_ (see below), expands the repertoire of electrophysiological features that can be presented by store-operated currents. These findings were rapidly reproduced in endothelial cells, in which it became quickly evident that STIM1 could associate with both Orai1 [38,39,40] and TRPC1 channels to mediate SOCE [41,86,87].

## 3. Ca^2+^-Selective vs. Non-Selective Cation Currents Activated by ER Ca^2+^ Store Depletion

As anticipated in Section 2.3, SOCE can be mediated by at least two distinct cation channels, which display clearly different electrophysiological features, i.e., I_CRAC_ and I_SOC_ (Figure 1) [88,89]. An additional I_CRAC_-like current, which is mediated by STIM1, TRPC1, TRPC4, and Orai1, has also been reported (Figure 1) [51,90,91].

### 3.1. STIM and Orai Proteins Mediate the I_CRAC_

The I_CRAC_ has been initially described and characterized in studies carried out in Jurkat T cells [92,93] and mast cells [32]. The I_CRAC_ displays a clearly discernible electrophysiological fingerprinting, including high Ca^2+^-selectivity (≈1000-fold more conducive for Ca^2+^ over Na^+^), low single-channel conductance (≈20 fS, i.e., 1000-fold lower than most ion channels), Na^+^ conduction upon removal of extracellular Ca^2+^ and Mg^2+^, inward-rectification (i.e., the property to preferentially conduct the current into the cell), fast Ca^2+^-dependent inactivation (CDI), and no clear E_rev_ up to +60 mV [32,93,94]. Large scale RNA interference (RNAi) screens—one using HeLa cells and four using Drosophila S2 cells—provided the straightforward evidence that the I_CRAC_ is mediated by the physical interaction between STIM proteins on the ER and Orai channels on the PM (Figure 1) [75,77,78,79].

Studies conducted on immune cells showed that, in the absence of extracellular stimulation, STIM1 and Orai1 proteins are uniformly distributed, respectively, in the ER and PM. Following agonist stimulation, InsP_3_-dependent reduction in [Ca^2+^]_ER_ stimulates STIM1 proteins to oligomerize and relocalize to ER-PM junctions, known as *puncta*, where peripheral ER cisternae come in close contact (10–25 nm) with the PM. Within these junctions, STIM1 binds to and gates hexamers of Orai1 channels through a trap-diffusion mechanism, thereby activating SOCE [94,95]. The mechanistic coupling between ER Ca^2+^ release and I_CRAC_/SOCE activation is strengthened by the tonic inhibition of InsP_3_Rs by inactivated STIM proteins [96]. Subsequent evidence confirmed that STIM1 and Orai1 support SOCE in virtually all cell types [52,74,94,95]. Orai1 presents a shorter splicing variant, known as Orai1β, which lacks 64 a.a. in the NH_2_-terminal tail but is still activated by ER Ca^2+^ depletion and can therefore mediate the I_CRAC_ [97]. However, Orai1β displays faster mobility in the PM [98] and a less pronounced CDI [97]. Orai1 presents two paralogue proteins, namely Orai2 and Orai3, which present different electrophysiological features [99,100]. For instance, the fast CDI is more pronounced in Orai3 as compared to Orai2, while Orai1 presents the slowest fast CDI among the three Orai isoforms [100]. Therefore, Orai1 proteins encode CRAC channels that exhibit larger I_CRAC_ and SOCE as compared to either Orai2 or Orai3 [95,100]. However, Orai2 and Ora3 can assemble with and negatively regulate Orai1 function [95]. Global knockout mice and CRISPR/Cas9 gene knockout in HEK-293 cells revealed that Orai1 can heteromerize with Orai2 and Orai3 in naive CRAC channels; these may, therefore, comprise different stoichiometric assortments of Orai isoforms and present distinct electrophysiological profiles as compared to Orai1 hexamers, thereby enhancing the bandwidth of intracellular Ca^2+^ signals [96,101].

STIM1 has a paralogue protein, namely STIM2 [76], which has a lower Ca^2+^ affinity (~500 µM vs. ~200 µM) and has long been thought to regulate resting Ca^2+^ entry and maintain [Ca^2+^]_ER_ [102]. In accord, a recent investigation showed that constitutive InsP_3_-mediated ER Ca^2+^ release promotes STIM2 pre-clustering at ER-PM junctions, thereby recruiting Orai1 channels and ensuring basal Ca^2+^ influx even in the absence of agonist stimulation [103,104]. Furthermore, STIM2 supports SOCE activation across the whole spectrum of agonist concentrations [96] and is indispensable to recruiting STIM1 to Orai1 channels also in response to weak agonist stimulation [103,104]. STIM2 presents alternative splice variants, such as STIM2.1, which inhibits SOCE by preventing Orai1 clustering in several cell types [105]; a long variant, known as STIM1L, which is mainly expressed in skeletal muscle [106]; and a short splice variant, termed STIM1B, which is located in neurons and results in slower I_CRAC_ kinetics and inactivation [107].

### 3.2. TRPC Channels Mediate the I_SOC_: Molecular Interaction with STIM1 and Functional Interplay with Orai1

The term I_SOC_ encompasses a variety of store-operated non-selective cation currents that present a different electrophysiological fingerprinting as compared to the highly Ca^2+^-selective CRAC channels (Figure 1) [88,108]. The I_SOC_ is featured by an E_rev_ ranging from 0 to ~+10 mV; slightly linear (or outwardly rectifying) IV relationship; permeability to K^+^ (outgoing) and Na^+^, Cs^+^, and Ca^2+^ (ingoing); and single-channel conductance in the pS range (~5–23 pS) [88,108]. These electrophysiological properties are quite different from those exhibited by CRAC channels but are strongly reminiscent of those presented by TRPC channels, which are all activated in response to PLC activation [108,109]. Six TRPC proteins (TRPC1, TRPC3, TRPC4, TRPC5, TRPC6, and TRPC7) have been identified in humans [108]. TRPC isoforms have been subdivided into four subgroups, i.e., TRPC1, TRPC2, TRPC4/TRPC5, and TRPC3/TRPC6/TRPC7, based upon their sequence homology and similarities in gating mechanisms [108,110]. The available data concur to support a major role for TRPC1 in mediating the I_SOC_ [88,89,111]. The Ambudkar group has provided solid evidence that, in human salivary gland (HSG) cells, InsP_3_-mediated ER Ca^2+^ depletion stimulates the I_CRAC_ via the formation of a STIM1-Orai1 complex, thereby triggering extracellular Ca^2+^ entry. The subsequent elevation in submembrane Ca^2+^ concentration promotes the exocytosis of TRPC1-containing vesicles, thereby inserting TRPC1 channels on the PM (Figure 2). Herein, TRPC1 is directly gated by STIM1 to mediate the I_SOC_ [85,112], and assembles into a signalling complex that already contains Orai1 and is located within ER-PM junctions [80,84]. The following studies confirmed that the dynamic STIM1/Orai1/TRPC1 ternary complex could be formed and mediate SOCE in other cell types, including human platelets [83] and megakaryocytes [113]. In agreement with these observations, Orai1-mediated extracellular Ca^2+^ entry can activate endogenous TRPC1 channels in HEK293 cells [114,115]. It is still disputed whether Orai1α and Orai1β can interchangeably serve to recruit and activate TRPC1 [97] or whether only Orai1α is required [116]. Conversely, STIM1L is more prone to gate TRPC1 than STIM1 [115]. The two distinct STIM1/Orai1 and STIM1/TRPC1 complexes could finely tune distinct Ca^2+^-dependent functions in HSG cells, in which the I_CRAC_ elicits a local Ca^2+^ signal that stimulates the nuclear translocation of the transcription factor, nuclear factor of activated T-cells (NFAT), while the I_SOC_ leads to a global increase in [Ca^2+^]_i_ that engages a different transcription factor, i.e., nuclear factor kappa-light-chain enhancer of activated B cells (NF-κB) [85,112].

The model presented above can be further implemented by recalling that STIM1 can also bind to and directly activate other TRPC isoforms, i.e., TRPC4 and TRPC5, and that STIM1-regulated TRPC channels can confer store dependency to their heteromeric partners, such as TRPC3 and TRPC6 [81,117]. Furthermore, TRPC1, TRPC4 and TRPC5 can assemble with each other into STIM1-gated heteromultimers, such as TRPC1/TRPC4 in human myoblasts [115,118] and TRPC1/TRPC4/TRPC5 in rat neonatal right ventricular cardiomyocytes [119] and mouse hippocampal neurons [120].

### 3.3. Do or Do Not Orai1 and TRPC1 form Heteromeric Channels? An Ongoing Controversy

The evidence that Orai1 and TRPC1 form a ternary complex including STIM1 in several cellular models led some authors to suggest that that these Ca^2+^-permeable channels do not only mediate distinct store-dependent currents but could also assemble into a heteromeric channel (Figure 1) [121,122,123]. According to this hypothesis, Orai1 and TRPC1 could both contribute to form the channel pore, thereby supporting a cation current with mixed properties between I_CRAC_ and I_SOC_ (e.g., E_rev_ ranging between 0 and +60 mV, larger Na^+^ selectivity, weak inward rectification or quasi-linearity). Alternatively, Orai1 has been suggested to serve as regulatory subunit by conferring store sensitivity to TRPC1 activation [83,124,125,126]. The prediction that TRPC1 and Orai1 might assemble and form heteromeric SOCs that exhibit electrophysiological features distinct from those mediated by TRPC1 and Orai1 alone has been largely discounted [108,121], with a few notable exceptions [123,124,125]. Nevertheless, in hypertrophied right ventricular cardiomyocytes, the integration of Orai1 in the STIM1/TRPC1/TRPC4 complex turns the I_SOC_ into an inwardly-rectifying non-selective cation current that reverses at around 0 mV [90]. These features resemble the mixed non-selective cation current that has been reported in aldosterone-treated rat left ventricular adult cardiomyocytes: this current is activated by depletion of the sarcoplasmic reticulum (SR), reverses at around 0 mV, presents a slight inward-rectification, and is significantly reduced by a dominant-negative Orai1 [119]. Interestingly, depletion of the SR Ca^2+^ store in these cells promoted the formation of a macromolecular complex including STIM1, TRPC1, TRPC4, and Orai1 [119]. Similarly, ER Ca^2+^ store depletion induced a non-selective cation current, which presented an inwardly-rectifying IV relationship, but with an E_rev_ ranging between–5 and 0 mV, in human macrophages [91] and in mouse Müller glia cells [127]. These currents were mediated by the synergistic activation of STIM1, Orai1, and TRPC1, although these studies did not document the formation of supermolecular ternary complex. A scrutiny of the literature reveals that additional store-operated currents with mixed electrophysiological features between I_SOC_ and I_CRAC_ have been reported. In human gingival keratinocytes [128] and bovine adrenal cells [129], SOCE involves the early activation of an I_CRAC_-like current that shows an E_rev_ ranging between +30 and +40 mV and is suppressed by the genetic deletion of TRPC4. TRPC4 presents a Ca^2+^/Na^+^ permeability ratio (P_Ca_/P_Na_) around 1.1, while the shape of the IV relationship depends on its physical interaction with other TRPC subunits. For instance, heterotetrameric TRPC4 channels present a doubly-rectifying IV relationship [130], while heteromeric combinations of TRPC1 with TRPC4 result in a non-selective cation current that reverses at around 0 mV and is outwardly-rectifying [131]. These observations strongly suggest that the I_CRAC_-like current recorded in human gingival keratinocytes [128] and bovine adrenal cells [129] could involve an endogenous subunit, e.g., Orai1, that had not been identified at that time and conferred to TRPC4 (maybe combined with TRPC1) the ability to inwardly-rectify and conduct Ca^2+^ better than Na^+^ [132].

## 4. The Endothelial I_CRAC_ Is Mediated by STIM1 and Orai1

The first evidence that SOCE is expressed in vascular endothelial cells has been provided in HUVECs by Ron Jacob and coworkers, who found a Ca^2+^ influx pathway that was “controlled by the degree of fullness of the internal (Ca^2+^) store”, i.e., the ER [133,134]. After this seminal discovery, the quest for the endothelial I_CRAC_ immediately started [49] and was based upon the electrophysiological fingerprinting of CRAC channels in mast cells [32].

### 4.1. Electrophysiological Properties of the Endothelial I_CRAC_

The first attempts to measure the I_CRAC_ by using the whole-cell patch-clamp technique in endothelial cell types failed whatever the chemical agonist employed to reduce the [Ca^2+^]_ER_ and the intracellular Ca^2+^-buffering conditions (i.e., introduction or not of Ca^2+^ buffers, such as EGTA or BAPTA, in the pipette solution), as shown in Table 1.

The first electrophysiological demonstration that vascular endothelial cells present the I_CRAC_ was provided by Cristina Fasolato and Bernd Nilius in 1998 [35]. The authors found that, in CPAE cells, depletion of the ER Ca^2+^ store evoked by tBHQ induced an inward current bearing the same electrophysiological properties of the I_CRAC_ recorded in Jurkat cells as the positive control (Table 2): (1) inward rectification; (2) E_rev_ > +40 mV; (3) small current density (−0.5 pA/pF at −80 mV, 20 mM Ca^2+^ in the external solution); (4) high Ca^2+^ selectivity and sensitivity to external La^3+^ (10 µM) and poor Ba^2+^ permeability; and (5) large permeability to monovalent cations in the presence of a divalent-free medium [35]. A decade later, these findings were confirmed by Mohamed Trebak and coworkers in HUVECs [38]. The I_CRAC_ recorded in these cells was evoked by intracellular dialysis of BAPTA or extracellular perfusion of thapsigargin displayed a rather small current density (around −0.3 pA/pF at −100 mV, 20 mM Ca^2+^ in the external solution), an inwardly-rectifying IV relationship, and an E_rev_ > +40 mV [38] (Table 2). Furthermore, this CRAC channel was also sensitive to extracellular lanthanides (10 µM Gd^3+^) and conducted large and fast-inactivating inward Na^+^ currents upon removal of extracellular divalent cations [38]. A subsequent study failed to measure a detectable I_CRAC_ in HUVECs [39], but the latter investigation exploited VEGF to stimulate the I_CRAC_ and this agonist could induce a smaller reduction in [Ca^2+^]_ER_, thereby resulting in the activation of a sub-pA ionic current [39]. Alternately, VEGF could target a sub-compartment of the ER that presents a weaker coupling to the CRAC channels on the PM as compared to thapsigargin [136]. These pioneering investigations demonstrated that vascular endothelial cells express an I_CRAC_ that displays similar electrophysiological properties to that recorded from mast cells.

### 4.2. STIM1 and Orai1 Mediate the Endothelial I_CRAC_

The I_CRAC_ recorded in HUVECs by Mohamed Trebak and coworkers was abolished by the genetic deletion of either STIM1 or Orai1 with specific small interfering RNAs (siSTIM1 and siOrai1, respectively) [38]. Conversely, this I_CRAC_ was not affected by the genetic suppression of TRPC1 and TRPC4 with a similar strategy [38]. As widely illustrated in other cell types, STIM1 showed a distributed microtubule-like distribution pattern in the presence of a replete ER, whereas it was stimulated by the depletion of ER Ca^2+^ to relocalize in subplasmalemmal clusters [39]. A study conducted on the HUVECs-derived cell line, EA.hy926, showed that intracellular Ca^2+^ overload may prevent STIM1-Orai1 interaction even when the ER Ca^2+^ pool is depleted [137]. However, mitochondrial Ca^2+^ uptake can prevent the Ca^2+^-dependent inhibition of STIM1-Orai1-dependent SOCE [138]. Intriguingly, the small current density of the I_CRAC_ in HUVECs was associated with the low basal levels of STIM1 expression: in accord, STIM1 overexpression produced an I_CRAC_ whose current density was similar to that measured in mast cells [38]. This feature might explain the difficulty in detecting a measurable I_CRAC_ in different endothelial cell types (Table 1) or in the same endothelial cell types (e.g., HUVECs), but provided by different sources [38,39]. Three additional mechanisms might contribute to reduce the current density of the endothelial I_CRAC_ below the resolution limit of conventional patch-clamp amplifiers (Table 1). First, ultrastructural analysis revealed that the distance between ER and PM in microvascular endothelial cells (87 nm) is significantly longer as compared to macrovascular endothelial cells (8 nm) [139]. If the minimum distance between the membranes that is required to accommodate STIM1- and Orai1-contaning *puncta* is 10–25 nm [52], this could explain why a true I_CRAC_ has never been recorded and SOCE is often lower [140] or even undetectable [141] in microvascular endothelial cells. Second, Orai2 may negatively modulate Orai1-mediated SOCE, as shown in the bovine brain capillary endothelial cell line, t-BBEC117 [142]. Therefore, changes in the expression levels of Orai2 (and, possibly, Orai3) in endothelial cells from different vascular districts and species could underlie the difficulties in recording the endothelial I_CRAC_. Third, Orai1 activity can ben modulated by multiple intracellular signalling pathways, which could thereby finely tune I_CRAC_ amplitude.

Agonist-evoked SOCE was compromised in HUVECs lacking STIM1 or Orai1 [38,39]. Therefore, SOCE was then exploited as readout of Orai1 activation to investigate the regulatory mechanisms of the endothelial I_CRAC_, which include: (1) tetraspanin Tspan18, which regulates Orai1 clustering and activity on the PM [22]; (2) SOCE-associated regulatory factor (SARAF) [143,144], which colocalizes with Orai1 and modulates the interaction between STIM1 and Orai1 during the early steps of SOCE activation [145]; (3) membrane cholesterol, which is required for STIM1-Orai1 interaction and whose reduction can significantly attenuate SOCE [146]; (4) redox signalling, which can directly inhibit Orai1 activity [147]; (5) the endocannabinoid N-arachidonoyl glycine (NAGly) [148] and the G protein-coupled estrogen receptor 1 (GPER) [149], which prevent STIM1-Orai1 interaction (6) STIM1 phosphorylation at Y361 via proline rich kinase 2 (Pyk2), which enables Orai1 recruitment to STIM1 *puncta* [150]; and (7) STIM1 phosphorylation by p38β mitogen-activated protein kinase (p38β MAPK), which inhibits SOCE activation [151]. Subtle changes in the extent of STIM1 or Orai1 modulation by any of these signalling pathways could affect I_CRAC_ activation and, therefore, favour or oppose I_CRAC_ recording depending not only upon the endothelial cell type, but also upon the experimental conditions.

### 4.3. The Role of STIM and Orai Proteins in Endothelial Function

The seminal discovery by the Trebak group [38] stimulated a flurry of studies that employed genetic tools or rather specific Orai1 inhibitors, such as low micromolar doses of trivalent cations and the pyrazole derivatives, YM-58483 (also known as BTP-2), Synta66 (S66), GSK-7975A (GSK) or Pyr6 [52,152,153], to unveil the endothelial functions modulated by the I_CRAC_ (Table 3).

The primary function fulfilled by STIM and Orai in endothelial cells is to refill the ER with Ca^2+^ (Table 3). A landmark study conducted on Ea.hy926 cells revealed that ER Ca^2+^ replenishment after InsP_3_-evoked ER Ca^2+^ mobilization is contributed by Orai1, as well as by the reverse-mode of the Na^+^/Ca^2+^ exchanger, and is physically supported by specific ER-PM junctions [154]. Furthermore, endothelial STIM1 and Orai1 support angiogenesis (Table 3) [24,30,155]. Genetic silencing of STIM1, STIM2, or Orai1 inhibited proliferation by causing cell cycle arrest at S and G2/M phase in HUVECs [38]. In the same endothelial cell type, genetic knockdown or pharmacological blockade with S66 or GSK abolished VEGF-induced extracellular Ca^2+^ entry, proliferation, and tube formation [39,143]. In addition, genetic silencing of Orai1 interfered with parathyroid hormone-induced HUVEC proliferation and migration [156] and with VEGF-induced Ca^2+^ entry and migration in human aortic endothelial cells (HAECs) [157]. In agreement with these in vitro findings, GSK inhibited aorta sprouting and the development of retinal vasculature in mice [143]. Intriguingly, RTK activation and PLCγ signalling were spatially restricted at the front of the migrating leader cell in HUVEC monolayers; herein, PLCγ promotes local InsP_3_-induced ER Ca^2+^ pulses that recruit STIM1 and selectively activate SOCE to accelerate cell-matrix adhesion and stimulate forward migration [60]. In addition, Orai1-mediated SOCE could promote angiogenesis by stimulating the nuclear translocation of NFAT (Table 3) [24,156].

Endothelial STIM1 and/or Orai1 can also control blood pressure by driving NO release (Table 3) [158]. Intriguingly, Lothar Blatter and his colleagues have provided evidence that the Ca^2+^ source responsible for the Ca^2+^-dependent recruitment of the eNOS is provided by SOCE rather than by InsP_3_-induced ER Ca^2+^ release [28,159]. An early investigation showed that genetic suppression of STIM1 inhibited thrombin-induced extracellular Ca^2+^ entry and NO release in porcine aortic endothelial cells [160]. A subsequent study generated a transgenic mouse model selectively lacking the endothelial STIM1, which showed impaired endothelium-dependent relaxation due to a robust reduction in acetylcholine-induced NO release [161]. These findings were confirmed by another study using endothelial cell-specific STIM1 knockout mice, which further showed a significant elevation in blood pressure [162]. Recent findings suggest that Orai1 mediates endothelium-dependent NO production also at the neurovascular unit, where NO is crucial to increasing cerebral blood flow in response to neuronal activity [4,163,164]. The pharmacological blockade of Orai1 inhibited NO release induced in human cerebrovascular endothelial cells by multiple neurotransmitters, such as glutamate [165,166], GABA [167,168], and acetylcholine [62], and neuromodulators, including histamine [169] and arachidonic acid [170]. Interestingly, Orai2 is the only isoform expressed in mouse cerebrovascular endothelial cells [21], which is consistent with the crucial role played by Orai2 in neuronal SOCE in mice [74]. The pharmacological inhibition of Orai2 also attenuated endothelium-derived NO production at the mouse neurovascular unit [21,171]. Orai3 has been shown to mediate VEGF-induced, arachidonic acid-dependent extracellular Ca^2+^ entry in HUVECs [172], but the role of this Orai isoform in endothelial SOCE is still unknown.

Interestingly, the Trebak group showed that STIM1 could control endothelial permeability in a SOCE-dependent manner. STIM1 was found to interact with the thrombin receptor, thereby recruiting the guanosine triphosphatase RhoA to stimulate myosin light chain phosphorylation and actin stress fiber formation [161]. A follow-up investigation confirmed that SOCE is also dispensable for histamine-induced formation of inter-endothelial gaps in microvascular endothelial cell monolayers [173]. In addition, the Hu group reported that, in HUVECs, a Leu-to-Pro substitution in the signal peptide enabled STIM1 to translocate to the nuclear membrane in response to a reduction in [Ca^2+^]_ER_ [174]. The mutated STIM1 was found to amplify the RyRs-mediated increase in nuclear Ca^2+^ concentration, thereby enhancing the subsequent cAMP responsive element binding protein activity, matrix metalloproteinase-2 (MMP-2) gene expression, and endothelial tube formation. It is noteworthy that the nuclear Ca^2+^ elevation was independent on SOCE activation [174]. These findings further expand the versatility of STIM1 signalling and place the question as to whether also Orai1 can signal in a non-canonical manner in endothelial cells.

**Table 3 ijms-24-03259-t003:** Endothelial functions modulated by the I_CRAC_.

Endothelial Cell Type	Evidence of I_CRAC_ Involvement	Function	Reference
Ea.hy926	Orai1^DN^	ER Ca^2+^ refilling	[154]
HUVECs	siSTIM1 and siOrai1, Orai^DN^, Synta66, BTP-2, GSK	Proliferation, cell cycle control (S and G2/M phase), tube formation	[38,39,143]
HUVECs	siSTIM1, STIM1^over^, BTP-2	Migration	[60]
Rat aortic endothelial cells	10–100 µM GSK (IC_50_ = 34.22 µM)	Aorta sprouting	[143]
Mouse retinal vasculature	2.6–31.8 mg/kg GSK (IC_50_ = 18.4 mg/kg)	Neovessel formation	[143]
Porcine aortic endothelial cells	siSTIM1	NO release	[160]
Endothelial cells from mesenteric resistance and thoracic arteries	STIM1^EC-/-^ mice	NO release and vasorelaxation	[161,162]
Mouse cerebrovascular endothelial cells	20 µM BTP-2, 10 µM La^3+^	NO release	[175]
Human cerebrovascular endothelial cells	10 µM Pyr6	NO release	[62,169]

Abbreviations: Orai1^DN^: expression of dominant negative Orai1; siOrai1: small interfering RNA selectively targeting Orai1; siSTIM1: small interfering RNA selectively targeting STIM1; STIM1^EC-/-^ mice: endothelial cell-specific knockout mice; STIM1^over^: STIM1 overexpression.

### 4.4. The Role of STIM and Orai Proteins in Endothelial Dysfunction

STIM1 and/or Orai1 drive endothelial cell activation during the inflammatory response to noxious stimuli and infection (Table 4). Early investigations revealed that the genetic silencing of STIM1 or Orai1 inhibited NFAT-dependent gene expression in HUVECs exposed to the inflammatory mediators, histamine [176], and tumor necrosis factor-α (TNF-α) [177]. Furthermore, genetic suppression of Orai1 attenuated TNF-α-induced expression of the adhesion molecules, VCAM-1 and ICAM-1, in mouse aorta and reduced the amount of pro-inflammatory cytokines, such as MCP-1, IL-6, and IL-8, in the serum (Table 4) [177]. Consistently, Lipopolysaccharide (LPS), which is derived from bacterial endotoxin, failed to induce intracellular Ca^2+^ oscillations, to stimulate the nuclear translocation of NFAT, and to increase vascular permeability in a transgenic mouse model selectively lacking the endothelial STIM1 (Table 4) [178]. These protective actions against systemic inflammation and acute lung injury (ALI) were mimicked by the small-molecule drug, BTP-2 [178]. In addition, BTP-2 alleviated endothelial hyperpermeability and lung oedema in a mouse model of ventilation-induced lung injury Orai1 may also exacerbate endothelial dysfunction under hyperglycaemic conditions, e.g., in diabetes [14]. The expression levels of STIM1-2 and Orai1-3 increased in HAECs during chronic exposure to high glucose (25 mM) and were also enhanced in the aorta of diabetic patients and of diabetic mice [179]. Furthermore, high glucose induced the up-regulation of Ora1-3 proteins and enhanced SOCE activity in human coronary artery endothelial cells (HCAECs), thereby inducing endothelial hyperpermeability and hyperproliferation [180]. In agreement with these observations, the genetic silencing of STIM1 and Orai1 alleviated the increase in permeability induced by the pro-inflammatory mediator, High-Mobility Group Box 1 protein (HMGB1), in Ea.hy926 monolayers [181], whereas a selective siOrai1 alleviated tunicamycin-induced ER stress and unfolded protein response, which are a feature of atherosclerosis, were shown to be alleviated by a selective siOrai1 in HUVECs [182]. Thus, the mechanisms by which chronic inflammation or hyperglycaemia lead to atherosclerosis involve an increase in endothelial permeability and proliferation that depend on STIM1 and Orai1 (Table 4) [183]. This evidence led to the hypothesis that Orai1 could contribute to inflammation-induced pulmonary endothelial cell injury during SARS-CoV-2 infection [184]. Moreover, Orai1-mediated extracellular Ca^2+^ entry mediates histamine- or thrombin-induced release of von Willebrand factor, thereby favouring platelet aggregation and thrombi formation in mouse models of deep vein thrombosis and ischemia-reperfusion cardiac injury [22]. As shown for angiogenesis and inflammation, Orai1-mediated extracellular Ca^2+^ influx stimulates the nuclear translocation of NFAT to induce the expression of the gene encoding for von Willebrand factor [22].

## 5. The Endothelial I_SOC_ Is Mediated by STIM1, TRPC1, and/or TRPC4

The search for the endothelial I_CRAC_ led to the discovery that ER Ca^2+^ depletion by stimuli that do not involve InsP_3_-induced Ca^2+^ release, i.e., the SERCA inhibitors CPA, thapsigargin, and tBHQ, induced either an I_SOC_ [47,187,188] or an I_CRAC_-like current [44,189]. In the present section, we discuss the evidence supporting the notion that the endothelial I_SOC_ is mediated by STIM1, TRPC1, and TRPC4, while in Section 6 we illustrate the findings indicating that the incorporation of Orai1 into this supermolecular complex turns the I_SOC_ into an I_CRAC_-like current.

### 5.1. Electrophysiological Properties of the Endothelial I_SOC_

The quest for the endothelial I_CRAC_ led to discovery of slightly different currents, which were activated by a reduction in [Ca^2+^]_ER_ but displayed the electrophysiological features of an I_SOC_ (Table 5) [49]. Depletion of the ER Ca^2+^ store with CPA in CPAE cells evoked a non-selective cation current that was carried by Na^+^, K^+^, Ca^2+^ and, possibly, the organic cation, tetraethylammonium, displayed a rather linear IV relationship and reversed close to 0 mV [187]. Furthermore, both CPA and the intracellular infusion of InsP_3_ caused the Ca^2+^-dependent recruitment of Ca^2+^-dependent K^+^ channels that induced endothelial hyperpolarization [187]. A membrane current bearing electrophysiological features similar to the I_SOC_ was then recorded in HUVECs [37,43,188,190] and in human and rat PAECs (hPAECs and rPAECs, respectively) [191,192].

### 5.2. STIM1, TRPC1, and TRPC4 Mediate the Endothelial I_SOC_

Groschner et al. were the first to suggest that the endothelial I_SOC_ was associated with an endogenous TRP channel likely belonging to the TRPC subfamily [37]. In agreement with this proposal, the Malik group reported that the intracellular infusion of InsP_3_ in HUVECs induced an I_SOC_ that was inhibited by a selective antibody targeting an extracellular epitope of TRPC1 and enhanced by TRPC1 overexpression [43]. Subsequently, CPA-evoked I_SOC_ was found to increase following TRPC4 overexpression in hPAECs [192].

The emerging evidence that TRPC1 and/or TRPC4 were both able to mediate SOCE and thereby control endothelial permeability, NO release, and gene expression [193,194,195,196,197], led to hypothesizing the assembly of a heterotetramer consisting of both subunits and recruitable by STIM1 in response to ER Ca^2+^ depletion (Figure 3). In agreement with this model, it was first revealed that TRPC1 and TRPC4 form a heterotetrameric channel in BAECs and mediate extracellular Ca^2+^ entry downstream of TKR activation. The evidence that neither subunit was phosphorylated suggested that the heterotetrameric complex was engaged upon ER Ca^2+^ depletion [198]. Subsequently, the Malik group showed that Ca^2+^ influx secondary to the physiological or pharmacological depletion of the ER Ca^2+^ store in mouse and human pulmonary artery microvascular endothelial cells (mPMECs and hPMECs, respectively) was inhibited by the genetic blockade of STIM1, TRPC1, and TRPC4 [41]. Coimmunoprecipitation studies revealed that STIM1 associates to TRPC1 and TRPC4 upon a reduction in [Ca^2+^]_ER_ [41] and suggested that the physical interaction between TRPC1 and TRPC4 is favoured by caveolin-1 (Figure 3) [195,199,200]. Intriguingly, caveolin-1 may also promote the association of TRPC1 and TRPC4 on the PM with InsP_3_Rs on the ER [195,199,201]. The co-localization of InsP_3_Rs, TRPC1, TRPC4, and STIM1 [202], in the same signalplex could accelerate SOCE activation by promoting the large drop in [Ca^2+^]_ER_ peripheral cisternae that is required to activate endothelial TRPC1 and TRPC4 channels, as recently shown for Orai1 in HeLa cells (Figure 3).

The endothelial TRPC1 and TRPC4 channels can be modulated by multiple signalling pathways [5,203,204], but only a few of them were shown to modulate both channels. For instance, TNF-alpha Receptor Ubiquitous Signaling and Scaffolding protein (TRUSS) and tumor necrosis factor receptor-1 (TNF-R1) interact with TRPC1 and TRPC4 to enhance SOCE and promote ER refilling after agonist evoked InsP_3_-induced Ca^2+^ mobilization [205]. Furthermore, protein kinase Cα phosphorylated TRPC1 and increased the I_SOC_ in HUVECs [43], whereas the monomeric GTP-binding protein RhoA favoured the association of InsP_3_Rs with TRPC1 after store depletion by promoting F-actin polymerization in hPAECs [201].

### 5.3. The Role of the I_SOC_ in Endothelial Function and Dysfunction

The Malik group has provided a major contribution to unveiling the pleiotropic role played by the endothelial I_SOC_ in endothelial cells (Table 6). The overexpression of TRPC1 increased vascular endothelial growth factor (VEGF)-induced extracellular Ca^2+^ influx and hyperpermeability in HUVECs and human dermal microvascular endothelial cells (HMECs) (Table 6) [193]. This effect was counteracted by angiopoietin-1, which interfered with the incorporation of InsP_3_Rs into the TRPC1-containing signalplex [193]. Moreover, genetic knockdown of TRPC1 also reduced angiotensin II-induced increase in [Ca^2+^]_i_ and permeability in HUVECs [206]. Likewise, thrombin-induced extracellular Ca^2+^ entry and increase in endothelial permeability was dramatically enhanced by TRPC1 up-regulation in hPAECs (Table 6) [201]. In agreement with these findings, genetic knockdown or deletion of, respectively, TRPC1 and TRPC4 inhibited the nuclear translocation of the pro-inflammatory transcription factor, NF-κB, in hPAECs (Table 6). The I_SOC_ was instrumental to activate AMPK and PKCδ, which are both required to activate NF-κB and increase endothelial permeability [207]. Similarly, tumor necrosis factor-α (TNF-α) induced I_SOC_ activation and thereby promoted endothelial hyperpermeability in EA.hy926 monolayers by causing the up-regulation of TRPC1, but not TRPC4, protein [208]. TNF-α is a primary inflammatory mediator that increases vascular permeability during the early stages of sepsis, and its stimulatory effect on endothelial cells can be counteracted by inhibiting the I_SOC_ with the chromogranin A (CGA)-derived fragment, CGA_47–66_ [208]. Finally, lead/malathion poisoning also increases the endothelial expression of TRPC1 and TRPC4 proteins and thereby causes an elevation in SOCE and permeability at the rat blood–brain barrier [209]. Therefore, these findings indicate that the I_SOC_ supports the endothelial Ca^2+^ signals that reorganize the cytoskeletal architecture to induce gaps between adjoining cells, disrupt the endothelial barrier and increase vascular permeability [210].

An additional process that is seemingly mediated by the endothelial I_SOC_ through NF-κB activation is cell survival during thrombin stimulation. In accord, HUVECs challenged with a prolonged exposure to thrombin activate the transcriptional pro-inflammatory program, but do not undergo apoptosis, due to concomitant NF-κB-dependent A20 expression [194]. More recent work suggested that TRPC1, as well as TRPC4, play an anti-apoptotic role also in rPAECs. Acute hypoxia resulted in the up-regulation of Galectin-3 (Gal-3), a member of the family of β-galactoside-binding lectins, which induced apoptosis by down-regulating both TRPC1 and TRPC4 proteins [211]. Therefore, the I_SOC_ could be effectively targeted to treat PAH.

Recent evidence suggested that endothelial TRPC1-containing channels could also control angiogenesis. For instance, endothelial TRPC1 supported VEGF-induced angiogenic sprouting of intersegmental vessels in zebrafish larvae [212], maintained the plateau phase of VEGF-induced extracellular Ca^2+^ entry in MAECs [213], induced wound healing in HMECs [214], and promoted vascular regrowth and recovery of myocardial function in a mouse model of acute myocardial infarction [215]. The glycosidase protein, Klotho, promotes TRPC1 association to VEGF receptor 2, thereby favouring TRPC1-mediated extracellular Ca^2+^ entry in response to InsP_3_-dependent reduction in [Ca^2+^]_ER_ [213]. In addition, VEGF could exploit TRPC4 to stimulate migration and tube formation in retinal microvascular endothelial cells [216]. Finally, the genetic silencing of TRPC1 and TRPC4 impaired proliferation and bidimensional tube formation in HUVECs [38,217].

**Table 6 ijms-24-03259-t006:** Endothelial functions modulated by the I_SOC_.

Endothelial Cell Type	Evidence of I_SOC_ Involvement	Function	Reference
HUVECs, HMECs	TRPC1^over^, anti-TRPC1 antibody, 1 µM La^3+^	VEGF-induced increase in endothelial permeability	[193]
HUVECs	siTRPC1	Angiotensin II-induced increase in endothelial permeability	[206]
HUVECs	siTRPC1	Thrombin-induced apoptosis	[194]
HUVECs	siTRPC1, TRPC1^over^	Ca^2+^- and TNFα-induced VCAM-1 upregulation and monocyte adhesion	[219]
hPAECs	TRPC1^over^	Thrombin-induced increase in endothelial permeability	[201]
hPAECs, mPAECs	siTRPC1, TRPC4^-/-^ mice	Nuclear translocation of NF-κB	[207]
Zebrafish	TRPC1^-/-^	Vascular development	[212]
HUVECs	siTRPC1, siTRPC4	Proliferation and tube formation	[38,217]
HRMECs	siTRPC4	VEGF-induced migration and tube formation	[216]
MAECs	siTRPC1	VEGF-induced Ca^2+^ entry	[213]
MCAECs	TRPC1^-/-^ mouse	Migration and tube formation in vitro and neovessel formation in vivo	[215]

Abbreviations: HMECs: human dermal microvascular endothelial cells; hPAECs: human pulmonary artery endothelial cells; HRMECs: human retinal microvascular endothelial cells; MAECs: mouse aortic endothelial cells; MCAECs: mouse coronary artery endothelial cells; mPAECs: mouse pulmonary artery endothelial cells; HUVECs: human umbilical vein endothelial cells; siTRPC1: small interfering RNA selectively targeting TRPC1; TRPC1^over^: TRPC1 overexpression.

Conversely, endothelium-dependent NO release and vasodilation were not affected in a transgenic mouse model devoid of TRPC1 [218]. This finding strongly suggests that TRPC1 is not tightly coupled to eNOS in vascular endothelial cells [218].

## 6. The I_CRAC_-Like Current: The Third Store-Operated Current in Endothelial Cells

Some of the earlier reports of the endothelial I_SOC_ described a membrane current that actually displayed intermediate electrophysiological features between I_SOC_ and I_CRAC_ (Table 7) [49]. Vaca and Kunze showed that, in BAECs, tBHQ elicited a non-selective cation current that displayed an inwardly-rectifying IV relationship, presented a single-channel conductance of 5 pS between −40 and −80 mV, reversed at ≈+6 mV, and could also be activated in response to physiological depletion of the InsP_3_-sensitive ER Ca^2+^ pool with ATP or bradykinin (Table 7) [44,189]. Subsequently, the Stevens group showed that thapsigargin activated an inwardly-rectifying cation current, which reversed between +30 mV and +40 mV and presented Ca^2+^-dependent inactivation, in rPAECs (Table 7) [69,140,220,221]. Intriguingly, the Na^+^ permeability of this store-dependent current was progressively reduced by a gradual increase in extracellular Ca^2+^ concentration [86]. This behaviour, which is known as anomalous mole fraction [52], is typical of Ca^2+^-selective channels, but the E_rev_ (+40 mV, more negative than the E_rev_ of a pure inward Ca^2+^ current) and the occurrence of a small outward current (which reflects cytosolic K^+^ efflux) indicate that this current is also contributed to by monovalent cations and is therefore non-selective. The Nilius group also showed that depleting the ER by dialyzing the cytosol with InsP_3_ through the patch-pipette and adding tBHQ to the bath elicited a store-operated current in MAECs (Table 7) [87]. This current reversed at ≈+40 mV and displayed a less robust inward rectification and lower Ca^2+^ permeability (P_Ca_/P_Na_ ≈ 160 vs. P_Ca_/P_Na_ ≈ 1000) as compared to the archetypal I_CRAC_ [52,222]. Finally, Frieden and colleagues found that, in EA.hy926 cells, thapsigargin evoked a store-dependent current that presented intermediate electrophysiological features between Ca^2+^-selective and non-selective cation channels (Table 7): (1) inwardly-rectifying IV relationship; (2) sensitivity to low micromolar doses of La^3+^; (3) anomalous mole fraction; and (4) rather negative E_rev_, i.e., ≈−7 mV [46]. In our opinion, these pieces of evidence strongly suggest that this endothelial store-operated current should not be coined either as I_SOC_ or as I_CRAC_, but rather as I_CRAC_-like [46], as suggested for the analogous store-dependent currents described in rat left ventricular adult cardiomyocytes [90,118], human gingival keratinocytes [128], and bovine adrenal cells [129].

### 6.1. STIM1, TRPC1, TRPC4, and Orai1 Mediate the Endothelial I_CRAC_-Like Current

The gating mechanisms and the electrophysiological features of the I_CRAC_-like current hint at a model according to which the channel pore is contributed to by both TRPC and Orai1 subunits, which assemble into a heteromeric complex that is informed about ER Ca^2+^ content by STIM1 [90,119,123]. In accord, thapsigargin-evoked I_CRAC_-like currents in rPAECs were significantly inhibited by the genetic knockdown of TRPC1 through a specific siRNA [86]. Subsequently, it was found that this I_CRAC_-like current requires spectrin-protein 4.1 interaction (Figure 4) [220] and protein 4.1 binding to TRPC4 (Figure 4) [69]. In parallel, the Nilius group discovered that the I_CRAC_-like current was suppressed upon genetic knockdown of TRPC4 with a selective siRNA and in MAECs isolated from TRPC4-deficient mice [87]. A landmark study to understand how the I_SOC_ can be turned into an I_CRAC_-like current was published in 2012, when the Stevens group exploited a novel Förster resonance energy transfer (FRET) approach to demonstrate that Orai1 could be incorporated into a protein 4.1-bound TRPC1/TRPC4 channel in rPAECs (Figure 4) [51]. They found that Orai1 is constitutively coupled to TRPC4 and can associate to TRPC1 in response to ER Ca^2+^ depletion [51]. This supermolecular channel complex includes 1 TRPC1 subunit and 2 TRPC4 subunits (Figure 4) [51,123], while the exact Orai1 stoichiometry remains to be investigated. Electrophysiological recordings revealed that genetic knockdown of Orai1 reduced the amplitude of the inward current at −80 mV, shifted the E_rev_ from +40 mV to +20 mV, reduced the Ca^2+^-dependent inactivation, and abolished the anomalous mole fraction behaviour [51]. A subsequent report from the same group confirmed that, although Orai1 is not strictly required to mediate Ca^2+^ entry in rPAECs, it favours the coupling between ER and PM channels during low ER Ca^2+^ depletion events and that it restricts the Na^+^ permeability of the I_CRAC_-like current [50]. Thus, as observed with the I_SOC_, Orai1 does not contribute to SOCE, but it finely regulates the cation permeability of CRAC-like channels: from non-selective (no Orai1 present) to Ca^2+^ (semi)-selective (Orai1 present in combination with TRPC channels). Finally, Cioffi and coworkers showed that genetic deletion of STIM1 reduced SOCE and that STIM1 phosphorylation upon ER Ca^2+^ release contributed to inactivate the I_CRAC_-like current in rPAECs [223]. The evidence that Orai1 can be incorporated into and confer CRAC-like features to TRPC1/TRPC4 heteromers can be crucial to unveiling the molecular architecture of the wide range of store-dependent currents, i.e., from truly to less Ca^2+^-selective, also recorded in other types of mammalian cells.

### 6.2. The Role of the I_CRAC_-Like Current in Endothelial Function

The role of the endothelial I_CRAC_-like current has been mainly investigated in rat pulmonary vasculature [51], in which it stimulates the development of intercellular gaps in extra-alveolar, but not intra-alveolar, endothelial cells [140,224]. This observation is consistent with the finding that, in rat PMECs (rPMECs), SOCE is lower as compared to rPAECs [140,225] and the I_CRAC_-like current is not detectable [140]. Intriguingly, blocking type 4 phosphodiesterase with rolipram increased intracellular levels of cAMP, abolished endothelial I_CRAC_-like currents in pulmonary artery, and promoted their occurrence in lung capillaries [140]. These findings concur with the notion that the cAMP-producing agonists, epinephrine, and isoproterenol, interfere with endothelial gap formation, and further reveal that the I_CRAC_-like current controls endothelial cell permeability [210]. Therefore, aberrant activation of the I_CRAC_-like current could sensitize alveolar endothelium to Ca^2+^-dependent openings that result in pulmonary edema [140,226]. It turns out that, at least in rodent pulmonary vasculature, the endothelial cell barrier could be finely tuned by endothelial Ca^2+^ signals supported by both the I_SOC_ (Section 4.3) and the I_CRAC_-like current. Conversely, the I_CRAC_ has been recorded in hPAECs (Table 2) and controls permeability in hPMECs (Table 4).

Endothelium-dependent NO production and vasodilation were impaired in the TRPC4 knockout mouse model lacking endothelial I_CRAC_-like currents [87]. In this view, it is worth of recalling that: (1) TRPC1 deletion does not generally affect endothelium-dependent NO production [218]; (2) eNOS is preferentially coupled with Orai1 [28,159]; and (3) Orai1 can be constitutively associated to TRPC4 [51]. Therefore, it is reasonable to speculate that, within the STIM1/TRPC1/TRPC4/Orai1 signalplex, the eNOS is selectively coupled to Orai1 via TRPC4.

## 7. The Endothelial SOCE Machinery: Open Questions

The evidence discussed so far, which derives from almost three decades of intense research, indicates that the endothelial SOCE can be mediated by at least three different ionic currents: (1) I_CRAC_, which is a Ca^2+^-selective current; (2) I_SOC_, which is a non-selective cation current; and (3) I_CRAC_-like, which bears intermediate electrophysiological features between I_CRAC_ and I_SOC_. Orai1 incorporation into the channel signalplex is likely to make the difference between a Ca^2+^-selective and a non-selective cation channel. It is, however, not surprising that many compelling issues regarding the molecular architecture and the role of endothelial SOCE in cardiovascular physiology and pathology remain to be elucidated.

First, may these three currents coexist in the same endothelial cell type? Pioneering work from the Ambudkar group showed that, following depletion of the ER Ca^2+^ pool, I_CRAC_ and I_SOC_ coexist in HSG cells, and that the I_CRAC_ is activated first to stimulate the Ca^2+^-dependent insertion of TRPC1 on the PM [85]. However, the Orai1-mediated I_CRAC_ is masked by the larger TRPC1-mediated I_SOC_ and can be easily detected only after genetic silencing of the latter conductance [85]. Intriguingly, patch-clamp whole-cell recordings revealed that store depletion first activated a tiny I_CRAC_-like current and then a larger I_SOC_ in human gingival keratinocytes [128,227]. This study showed that genetic deletion of TRPC4, a core component of the I_CRAC_-like current, also suppressed the subsequent I_SOC_ [128]. These findings could help explaining how Orai1 and TRPC1/TRPC4 interact in endothelial cells and some discrepancies between different studies. For instance, both I_CRAC_ [38] and I_SOC_ [37,43,188,190] were recorded in HUVECs. However, Abdullaev et al. did not report about a large non-selective cation current, although they were able to record the nearly unmeasurable endothelial I_CRAC_ [38]. This investigation exploited a Cs^+^-based intracellular solution, which is routinely employed to measure the I_CRAC_, while TRPC1 may display a poor permeability to Cs^+^ [228]. Conversely, the pipette solution employed to record the I_SOC_ was based upon K^+^ [37] or Na^+^ [43,193], which readily permeate TRPC1 and can generate large outward currents. In addition, Abdullaev et al. dialyzed the cytosol with 20 mM BAPTA to record the I_CRAC_ [38], which might have inadvertently prevented the local Ca^2+^ entry through Orai1 from inducing TRPC1 exocytosis on the PM. Therefore, these authors might have found the ideal conditions to record the I_CRAC_ long before the role of Orai1 in stimulating the I_SOC_ was identified [85]. In line with this hypothesis, VEGF-induced I_SOC_ in HUVECs was inhibited by 1 µM La^3+^ [193], which selectively targets Orai1 rather than TRPC1 [52,152,229]. In our opinion, these data collectively support the notion that I_CRAC_ and I_SOC_ coexist in HUVECs. Another popular model to investigate the role of SOCE in endothelial physiopathology is represented by pulmonary endothelium. Two independent studies showed that both the I_CRAC_ [38] and the I_SOC_ [192] are present in hPAECs: while the former is mediated by STIM1/Orai1 [38], studies conducted on both hPAECs [192] and hPMECs [41] suggest that the latter is mediated by STIM1, TRPC1, and TRPC4. Like in HUVECs, it is yet to assess whether the I_SOC_ recorded in hPAECs requires previous Orai1-mediated Ca^2+^ entry. Intriguingly, this I_SOC_ is again sensitive to 1 µM La^3+^ [192]. Intriguingly, while SOCE in mPMECs is sensitive to the genetic blockade of STIM1, TRPC1, and TRPC4, but not of Orai1 (see Section 5.2), SOCE in rPAECs is due to an I_CRAC_-like current that requires the assembly of a STIM1/TRPC1/TRPC4/Orai1 supermolecular complex (see Section 6.1). Therefore, the molecular machinery responsible for SOCE activation in human and rodent lung endothelia could be different, and a further divergence can be observed between mouse and rat PAECs. Nevertheless, the evidence that the small molecule inhibitor BTP2, which is commonly employed to inhibit Orai1, halts sepsis- and ventilation-induced ALI suggests that Orai1 can somehow interact with TRPC1 and TRPC4 also in mice pulmonary endothelial cells [178,183]. In our opinion, this finding raises the question as to whether Orai1 can be incorporated into the STIM1/TRPC1/TRPC4 signalplex also in mPMECs. This hypothesis is supported by two pieces of evidence: (1) genetic deletion of Orai1 does not remarkably affect SOCE amplitude upon activation of endothelial I_CRAC_-like currents (this feature would explain why SOCE is unaffected in mPMECs transfected with a siOrai1) [50]; and (2) extracellular Na^+^ influx through CRAC-like channels can also regulate endothelial function (this feature would explain the beneficial effect of the systemic infusion of BTP-2 against LPS-induced pulmonary damage in mice) [178,230]. Electrophysiological measurements of the store-operated currents in mPMECs, which are still lacking, will help definitively in solving this issue.

Second, why have only a few studies identified a clear role of TRPC4 in either the I_SOC_ or the I_CRAC_-like current? Is the endothelial I_SOC_ strictly dependent on TRPC4 or can it be contributed to only by TRPC1? Ahmed et al. reported that the application of a specific antibody selectively targeting an extracellular epitope of TRPC1 significantly reduced, but not fully suppressed, InsP_3_-evoked I_SOC_ in HUVECs [43]. The remaining current could, therefore, be due to an incomplete block of TRPC1 because of its physical association with TRPC4 within the channel signalplex. It should, however, be pointed out that TRPC4 expression on the PM may depend on endothelial cell confluence [231]. The Groschner group has convincingly shown that TRPC4 is recruited into the PM and mediates extracellular Ca^2+^ entry only in sub-confluent, proliferating vascular endothelial cells. Conversely, TRPC4 did not support Ca^2+^ influx in either isolated or fully confluent endothelial cells [232]. Under such conditions, therefore, TRPC-based store-operated currents are likely to be contributed to only by STIM1 and TRPC1.

Third, what are reasons for such molecular heterogeneity in SOCE machinery in vascular endothelial cells? It has been suggested that Orai1 and TRPC1/TRPC4 channels generate spatially separated Ca^2+^ microdomains that engage distinct Ca^2+^-dependent effectors, such as NFAT and NF-κB (Figure 5) [85,121]. In vascular endothelial cells, the I_CRAC_ primarily regulates ER Ca^2+^ refilling, angiogenesis, and NO release, and platelet aggregation (see Section 4.3), but may also support platelet aggregation and thrombi formation as well as systemic inflammation and pulmonary vascular hyperpermeability (see Section 4.4). Conversely, the I_SOC_ and the I_CRAC_-like current have long been associated with the fine regulation of endothelial permeability (see Section 5.3 and Section 6.2). Intriguingly, I_CRAC_ and I_SOC_ engage two different transcription factors, i.e., NFAT and NF-κB, respectively, to stimulate the gene expression pro-inflammatory program (Figure 5) [178,194,207]. Recent findings, however, showed that the I_SOC_ could also support the angiogenic activity (Figure 5 and Table 6) and that the I_CRAC_-like can induce vasorelaxation via NO release [87]. These findings suggest that Orai1 and TRPC1/TRPC4 channels could converge on the same endothelial processes (e.g., permeability and vasorelaxation) either via distinct (permeability, NFAT vs. NF-κB) or similar (vasorelaxation, eNOS) Ca^2+^-dependent effectors. The pure Ca^2+^-selective I_CRAC_ could then regulate additional Ca^2+^-dependent processes, such as ER Ca^2+^ refilling and van Willebrand factor release.

Fourth, does endothelial SOCE cooperate with perinuclear InsP_3_Rs via the CICR mechanism to stimulate an increase in perinuclear or nuclear Ca^2+^? Early studies demonstrated that growth factors induced oscillations in nuclear Ca^2+^ concentration in both micro- [233] and macro-vascular [234] endothelial cells and that these oscillations were abrogated by the pharmacological blockade of SOCE. Future work could assess whether the endothelial SOCE stimulates perinuclear Ca^2+^ release through InsP_3_Rs, thereby inducing the nuclear translocation of NFAT and/or NF-κB, and whether the nuclear Ca^2+^ oscillations sustained by SOCE recruit nuclear localized transcriptional regulators, such as DREAM and myocyte enhancer factor 2C.

Fifth, do other Ca^2+^-entry pathways play a role in endothelial SOCE? This question does not simply arise from the evidence that other TRPC isoforms may support SOCE in heterologous expression studies, but from landmark studies conducted in native endothelial cells. It has been shown that, in EA.hy926 cells, depleting the ER Ca^2+^ store with thapsigargin activated both the STIM1/Orai1 machinery and TRPC3-mediated Ca^2+^ entry [40]. In accord, depletion of the ER Ca^2+^ content was found to stimulate DAG production via PLC activation, which could be explained by the known Ca^2+^-dependence of PLC activity; DAG, in turn, stimulated the novel PKCη to activate Src kinase and phosphorylate TRPC3, thereby promoting further Ca^2+^ entry [40]. In addition, TRPC1 can assemble into a heteromeric channel with TRPV4 in rat mesenteric artery endothelial cells [235], and a reduction in [Ca^2+^]_ER_ can favour the translocation of TRPV4/TRPC1 complexes into the PM, where they are activated by STIM1 and mediate a large I_SOC_ [236]. These findings were confirmed in MAECs and HUVECs [42], and it was further shown that this novel mode of endothelial SOCE could induce vasorelaxation via NO release in resistance arterioles [237]. It would be tempting to speculate that TRPC1 associates to TRPV4 when TRPC4 is not (or only barely) expressed on the PM, but this hypothesis remains to be experimentally probed.

Sixth, which store-dependent current(s) mediate SOCE in ECFCs? SOCE supports ECFC proliferation, migration, and tube formation both in vitro [25,61,238,239,240,241,242] and in vivo [26,27]. Electrophysiological recordings revealed that a store-operated current is not detectable in ECFCs, whereas Ca^2+^ imaging revealed that their SOCE is impaired by the genetic knockdown of STIM1, Orai1, and TRPC1 [30,39,239]. If SOCE is supported by TRPC1, but the depletion of the ER Ca^2+^ content does not lead to the activation of a measurable membrane current, the latter could be mediated by a channel pore that is contributed to by both Orai1 and TRPC1. Future work will have to assess whether STIM1, Orai1, TRPC1, and, possibly, TRPC4 assemble within the same ion channel signalplex and mediate an I_CRAC_-like current in ECFCs. The evidence that, in ECFCs, SOCE activation results in NF-κB activation [25,238] strongly suggests that the tight coupling between the Ca^2+^-sensitive transcription factor and the SOCE machinery is provided by TRPC1.

## 8. Conclusions

SOCE maintains the endothelial Ca^2+^ response to extracellular stimulation and thereby regulates a plethora of endothelial functions, including angiogenesis, permeability, platelet aggregation, and NO release. Three distinct store-operated currents can mediate SOCE in vascular endothelial cells: (1) the Ca^2+^-selective I_CRAC_, which is mediated by STIM1 and Orai1; (2) the non-selective I_SOC_, which is mediated by STIM1, TRPC1, and TRPC4; and (3) the moderately Ca^2+^-selective, I_CRAC_-like current, which is mediated by STIM1, TRPC1, TRPC4, and Orai1. The incorporation of Orai1 into the STIM1/TRPC1/TRPC4 complex, therefore, turns the non-selective I_SOC_ into a Ca^2+^ (semi)-selective current. As highlighted in Section 7, many questions remain open. In our opinion, the most challenging issues that need to be urgently addressed to move a significant step forward are the following: (1) may these three Ca^2+^-permeable currents coexist in the same endothelial cell type or does their endothelial distribution change along the vascular network and/or from one species to another? (2) are the I_CRAC_, I_SOC,_ and I_CRAC_-like currents coupled to distinct Ca^2+^-dependent decoders, and do they modulate different endothelial functions? (3) does Orai1-mediated extracellular Ca^2+^ entry result in TRPC1 translocation to the plasma membrane also in endothelial cells? Exploiting the novel technologies offered by both molecular biology (e.g., CRISPR/Cas9, single-cell RNA sequencing, endothelial-specific GCaMP3 mice) and imaging techniques (e.g., multiphoton microscopy to detect submembrane Ca^2+^ entry events) will help in solving these issues. Understanding the molecular architecture of this mechanism is indispensable in efficiently targeting endothelial SOCE in patients affected by life-threatening cardiovascular disorders, such as atherosclerosis, cerebrovascular dysfunction, sepsis-induced systemic inflammation, and PAH.

## Figures and Tables

**Figure 1 ijms-24-03259-f001:**
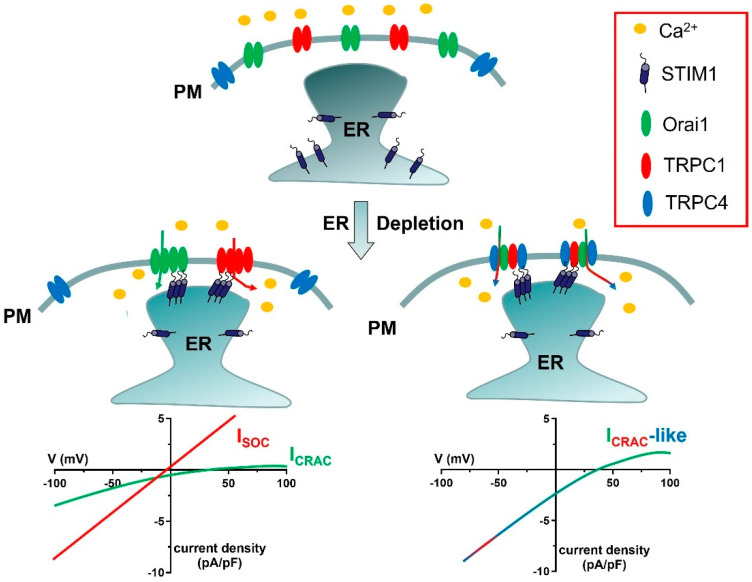
Proposed models for I_CRAC_, I_SOC_, and I_CRAC_−like currents. In the absence of extracellular stimulation, STIM1 is homogeneously distributed within ER cisternae, whereas Orai1, TRPC1, and TRPC4 are located on the PM. Upon depletion of the ER Ca^2+^ store, STIM1 aggregates and translocates in close apposition to the PM, thereby recruiting Orai1 hexamers into spatially confined puncta and activating the I_CRAC_. Orai1−mediated extracellular Ca^2+^ entry can cause TRPC1 insertion into the plasma membrane (shown in Figure 2), thereby enabling TRPC1 activation by STIM1 and activating the I_SOC_. Finally, STIM1 can determine the assembly of a complex ion channel signalplex consisting also of Orai1, TRPC1, and TRPC4 and responsible for the development of I_CRAC_−like currents. As explained in Section 6.1, this supermolecular channel complex includes 1 TRPC1 subunit and 2 TRPC4 subunits. The lower current density of the I_CRAC_ as compared to the I_SOC_ and the I_CRAC_-like current reflects the single-channel conductance of Orai1 channels, which is 1000-fold lower as compared to TRPC channels. The current density is defined by the ratio between the magnitude of an ion current, in pA, and the cell membrane capacitance, in pF.

**Figure 2 ijms-24-03259-f002:**
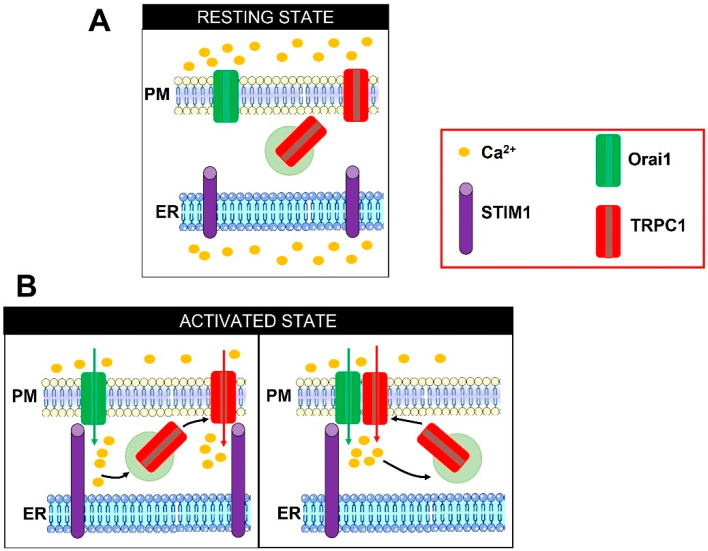
Illustrations describing the two proposed models of Orai1-dependent TRPC1 activation. In the absence of extracellular stimulation, TRPC1 is located both on the PM and on submembrane vesicles (**A**). Depletion of the ER Ca^2+^ store prompts STIM1 to oligomerize, extend the cytosolic COOH-terminal domain towards the PM, translocate to ER-PM junction, and physically engage Orai1 to mediate the I_CRAC_ (**B**, **left panel**). The following influx of Ca^2+^ can induce the exocytosis of TRPC1-containing vesicles. TRPC1 is inserted into the PM in close apposition to Orai1 and is thereafter activated by STIM1 to mediate the I_SOC_ (**B**, **left panel**). Alternately, TRPC1 can physically interact with Orai1 and indirectly become store-operated (**B**, **right panel**). Adapted from [88].

**Figure 3 ijms-24-03259-f003:**
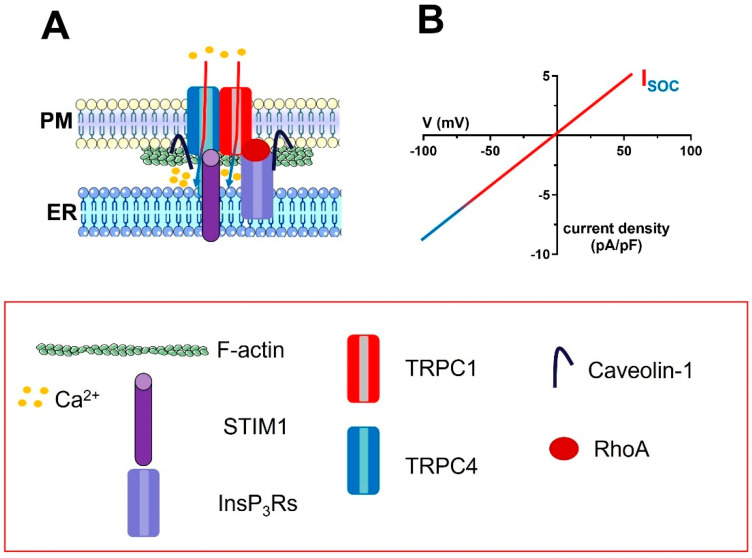
The molecular architecture of the I_SOC_ in vascular endothelial cells. (**A**), the endothelial I_SOC_ is mediated by a complex ion channel signalplex that is located on plasmalemmal caveolae. The ion channel pore is contributed by TRPC1 and TRPC4 channels, which are informed about changes in [Ca^2+^]_ER_ by STIM1. The signalling microdomain is enriched with InsP_3_Rs, which interact with the TRPC1 and TRPC4 subunits via caveolin-1. For sake of clarity, only the interaction between InsP_3_Rs and TRPC1 has been shown. The monomeric GTP−binding protein, RhoA, also supports the interaction between InsP_3_Rs and TRPC1 via F−actin polymerization (two bundles of F-actin were drawn beneath the plasma membrane). (**B**), the IV relationship of the endothelial I_SOC_.

**Figure 4 ijms-24-03259-f004:**
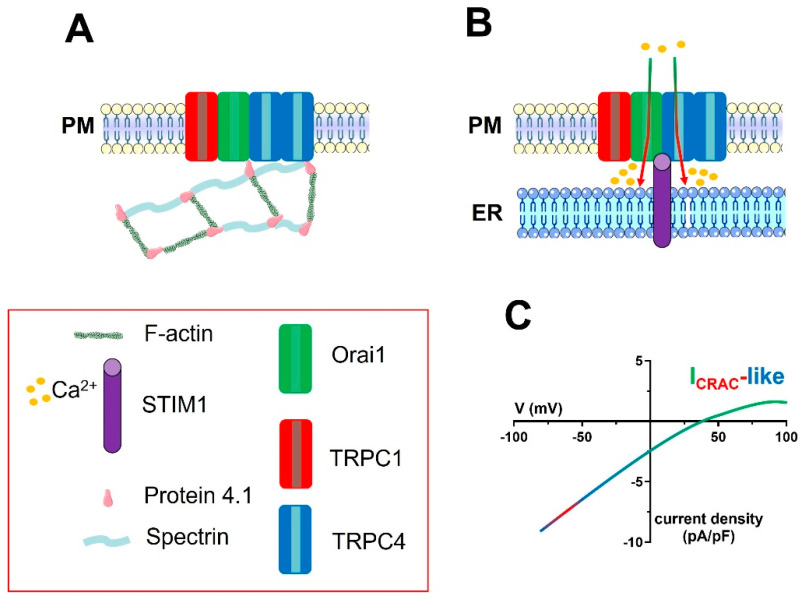
The molecular architecture of the I_CRAC_-like channel in vascular endothelial cells. A series of studies carried out on rPAECs demonstrated that the ion channel signalplex mediating the endothelial I_CRAC_−like currents is contributed to by one TRPC1 subunit and two TRPC4 subunits, as well as by Orai1, and is located within caveolae. TRPC4 is physically associated with both Orai1 and the actin-binding protein, protein 4.1. The latter, in turn, must be associated with the spectrin membrane skeleton (**A**). A reduction in [Ca^2+^]_ER_ (not shown) causes the STIM1-dependent activation of the ion channel signalplex on the PM (**B**). STIM1 is likely to physically interact with Orai1, TRPC1, and TRPC4, but this hypothesis remains to be experimentally probed. Orai1 incorporation into the STIM1/TRPC1/TRPC4 complex determines the Ca^2+^−selectivity of the store-operated current, which can therefore be defined as I_CRAC_-like (**C**). For sake of clarity, the spectrin−F-actin network beneath the plasma membrane has not been shown in Panel B.

**Figure 5 ijms-24-03259-f005:**
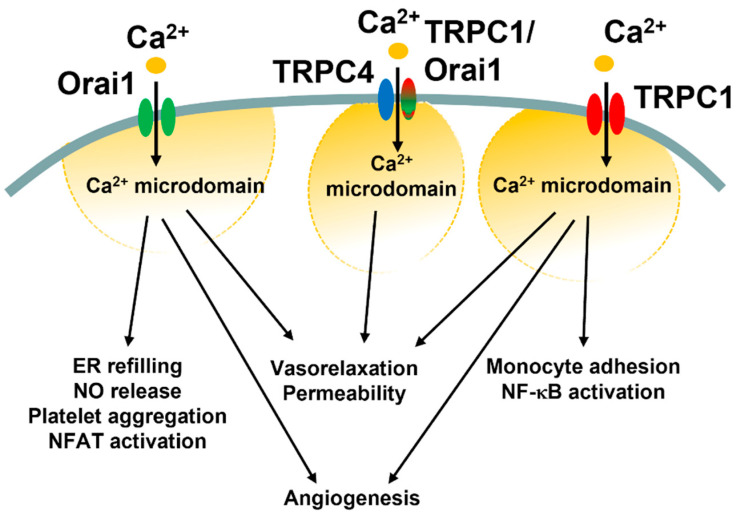
Endothelial functions regulated by Ca^2+^ entry through the diverse SOCE mechanisms. This illustration summarizes the different endothelial functions regulated by the ICRAC (Orai1 in green), I_CRAC_-like currents (TRPC4 in blue plus TRPC1/Orai1 in red/green), and I_SOC_ (TRPC1 in red).

**Table 1 ijms-24-03259-t001:** Unsuccessful attempts to measure the endothelial I_CRAC_.

Endothelial Cell Type	Measured I_CRAC_	Agonist Used to Reduce the [Ca^2+^]_ER_	Intracellular Ca^2+^ Buffering	Reference
HUVECs	No	Extracellular thapsigargin	0.1 mM EGTA, no CaCl_2_	[135]
HUVECs	No	Intracellular InsP_3_ or extracellular thapsigargin	0.1 mM EGTA, no CaCl_2_; 10 mM BAPTA no CaCl_2_	[34]
CPAE cells	No	10 mM EGTA/BAPTA	10 mM EGTA; 10 mM EGTA or 10 mM BAPTA	[33]
		InsP3 + 10 mM EGTA/BAPTA		
		Thapsigargin		
		InsP_3_ + Thapsigargin +10 mM EGTA		

Abbreviations: CPAE cells: calf pulmonary artery endothelial cells; HUVECs: human umbilical vein endothelial cells.

**Table 2 ijms-24-03259-t002:** Electrophysiological features of the endothelial I_CRAC_.

Endothelial Cell Type	I_CRAC_: Electrophysiological and Pharmacological Features	Agonist Used to Reduce the [Ca^2+^]_ER_	Intracellular Ca^2+^ Buffering	Reference
CPAE cells	IR, E_rev_ > +40 mV, depotentiation of the Na^+^ currents under DVF conditions, inhibited by 10 µM La^3+^	Intracellular InsP_3_	12 mM BAPTA	[35]
HUVECs, hPAECs	IR, E_rev_ > +40 mV, depotentiation of the Na^+^ currents under DVF conditions, inhibited by 10 µM Gd^3+^, activated by 5 µM 2-APB	Intracellular dialysis with BAPTA or extracellular thapsigargin	20 mM BAPTA or [Ca^2+^]_i_ buffered at 98 nm/L for thapsigargin-evoked currents	[38]

Abbreviations: 2-APB: 2-Aminoethoxydiphenyl borate; CPAE cells: calf pulmonary artery endothelial cells; DVF: divalent-free; IR: inward rectification; E_rev_: reversal potential; hPAECs: human pulmonary artery endothelial cells; HUVECs: human umbilical vein endothelial cells.

**Table 4 ijms-24-03259-t004:** Role of the I_CRAC_ in endothelial dysfunction.

Endothelial Cell Type	Evidence of I_CRAC_ Involvement	Function	Reference
HUVECs	siOrai1, Orai1^over^	ICAM-1 and VCAM-1 expression, monocyte adhesion	[177]
Mouse aorta and lungs	Orai1^over^	Expression of ICAM-1 and VCAM-1 and of pro-inflammatory cytokines (IL-6, IL-8, MCP-1 and E-selectin) in the aorta, lung inflammation	[177]
HUVECs	siSTIM1, siOrai1	LPS-induced apoptosis	[185,186]
mPAECs	STIM1^EC-/-^ mice, BTP-2 (1 mg/kg for in vivo experiments; 5 µM BTP-2 for in vitro experiments)	LPS-induced leukocyte infiltration, increase in pro-inflammatory cytokines, endothelial cell death, vascular leakage and pulmonary edema	[178]
hPMECs	BTP-2 (1 mg/kg for in vivo experiments; 20 µM BTP-2 for in vitro experiments)	Cyclic stretching-induced increase in endothelial permeability Ventilation-induced increase in pulmonary permeability	[183]
EA.hy926	siSTIM1, siOrai1, 1–20 µM SKF-96365 and 50–70 µM 2-APB	HMGB1-increase in permeability	[181]
HUVECs	siOrai1	von Willebrand factor release	[22]

Abbreviations: ALI: acute lung injury; hPMECs: human pulmonary microvascular endothelial cells; HMGB1: High-mobility group box 1 protein; HUVECs: human umbilical vein endothelial cells; LPS: lipopolysaccharide; mPAECs: mouse pulmonary artery endothelial cells; rPMECs: rat pulmonary microvascular endothelial cells; siOrai1: small interfering RNA selectively targeting Orai1; siSTIM1: small interfering RNA selectively targeting STIM1; STIM1^EC-/-^ mice: endothelial cell-specific knockout mice; STIM1^over^: STIM1 overexpression.

**Table 5 ijms-24-03259-t005:** Electrophysiological features of the endothelial I_SOC_.

Endothelial Cell Type	I_SOC_: Electrophysiological and Pharmacological Features	Agonist Used to Reduce the [Ca^2+^]_ER_	Intracellular Ca^2+^ Buffering	Reference
hPAECs	Slightly DR, E_rev_ ≈ 0 mV, inhibited by 1 µM La^3+^	CPA	1.15 mM EGTA, no CaCl_2_	[192]
CPAE cells	Linear IV, E_rev_ ≈ 0 mV, permeability to Na^+^, K^+^, Ca^2+^, inhibited by 50 µM La^3+^	CPA	0.3 mM CaCl_2_, 2.2 mM EGTA	[187]
HUVECs	Single-channel conductance: 28 pS (140 mM KCl in the bath and the pipette)	CPA	Cell-attached single channel study	[190]
HUVECs	Linear IV, E_rev_ ≈ 0 mV, permeability to Na^+^, K^+^, Ca^2+^, inhibited by 50 µM La^3+^	Intracellular infusion of InsP_3_	10 mM BAPTA, no CaCl_2_	[37,188]
HUVECs	Linear IV, E_rev_ ≈ 0 mV, inhibited by 1 µM La^3+^	Intracellular infusion of 3-deoxy-3-fluoro-D-myo-InsP_3_ (metabolism-resistant InsP_3_) and/or thapsigargin	12 mM BAPTA, no CaCl_2_	[43,193]

Abbreviations: CPAE cells: calf pulmonary artery endothelial cells; HUVECs: human umbilical vein endothelial cells.

**Table 7 ijms-24-03259-t007:** Electrophysiological features of the endothelial I_CRAC_-like current.

Endothelial Cell Type	I_SOC_: Electrophysiological and Pharmacological Features	Agonist Used to Reduce the [Ca^2+^]_ER_	Intracellular Ca^2+^ Buffering	Reference
BAECs	IR, E_rev_ ≈ +6 mV, permeable to Na^+^, Ca^2+^ and Ba^2+^, P_Ca_/P_Na_ > 10, single-channel conductance: 5 pS (symmetrical Na_2_SO_4_), inhibited by 200 µM La^3+^	tBHQ	2 mM CaCl_2_, 5 mM BAPTA	[44,189]
rPAECs	Slight IR, E_rev_ ≈ +30/+40 mV, anomalous mole fraction behaviour, inhibited by 50 µM La^3+^	Extracellular application of thapsigargin	[Ca^2+^]_i_ buffered at 100 nm/L	[51,69,86,140,220,221]
MAECs	IR, E_rev_ ≈ +40 mV, Na^+^ permeability anomalous mole fraction behaviour, P_Ca_/P_Na_ ≈ 160, inhibited by 1 µM La^3+^	Intracellular dialysis of InsP_3_ and extracellular application of tBHQ	12 mM BAPTA	[87]
EA.hy926	IR, Er_ev_ ≈ −7 mV, anomalous mole fraction behaviour, Ca^2+^-dependent inactivation, inhibited by 10 µM La^3+^	Extracellular application of thapsigargin	No CaCl_2_, no BAPTA/EGTA	[46]

Abbreviations: BAECs: bovine aortic endothelial cells; IR: inward rectification; MAECs: mouse aortic endothelial cells; P_Ca_/P_Na_: Ca^2+^/Na^+^ permeability ratio; rPAECs: rat pulmonary artery endothelial cells.

## Data Availability

Not applicable.

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
