# Peer review of "The Molecular Heterogeneity of Store-Operated Ca^2+^ Entry in Vascular Endothelial Cells: The Different roles of Orai1 and TRPC1/TRPC4 Channels in the Transition from Ca^2+^-Selective to Non-Selective Cation Currents"

_ijms, 2023, doi:10.3390/ijms24043259_

Round 1
Reviewer 1 Report
In the present review article ijms-2157829, entitled: “The molecular heterogeneity of store-operated Ca2+ entry in vascular endothelial cells: from Ca2+-selective to non-selective cation channels. Does Orai1 make the difference?”, Moccia and coworkers examined the molecular mechanisms that mediate SOCE in vascular endothelial cells.
In my opinion, this topic deserves careful consideration since, on the one hand, SOCE prolongs the Ca2+ response to virtually all endothelial agonists and, on the other hand, its molecular makeup in vascular endothelium remains elusive. Indeed, although all recent investigations focused on the involvement of STIM and Orai proteins, early work clearly revealed the contribution of TRPC1 and TRPC4.
In this article, the authors carry out an extensive survey of relevant literature and provide solid evidence to support their view, according to which three distinct currents can mediate SOCE in vascular endothelial cells: 1) the Ca2+-selective Ca2+-release activated Ca2+ current (ICRAC), which is mediated by STIM1 and Orai1; 2) the store-operated non-selective current (ISOC), which is mediated by STIM1, TRPC1, and TRPC4; and 3) the moderately Ca2+-selective, ICRAC-like current, which is mediated by STIM1, TRPC1, TRPC4, and Orai1.
This long manuscript is full of information and extensively evaluates the contribution to this field of research.
Nevertheless, a number of issues should be addressed before it is suitable for publication:
1) The Authors do not mention that Orai3 could mediate arachidonate-evoked Ca2+ entry in vascular endothelial cells (PMID: 26160956). This information should be added to Section 3.3.
2) STIM1 can regulate endothelial permeability independently on Ca2+ signalling (PMID: 23512989; PMID: 24965067). This interesting information should also be added to Section 3.3.
3) In Table 3, LaCl3 should be replaced by La3+ as elsewhere in the manuscript.
4) The concentration of each SOCE inhibitor reported in the tables should be indicated. Currently, only La3+ and Gd3+ concentrations are indicated.
5) There is a typo in line 620: “TRPC4to stimulate”.
6) Line 836: Does the evidence that SOCE is required for VEGF to activate NF-kB suggests that TRPC1 contributes to the channel pore in circulating ECFCs?
7) The Tables are very useful. I recommend to add one further figure to elucidate the role of Orai1 and TRPC1/TRPC4 in endothelial cell physiology.
Reviewer 2 Report
This manuscript is reviewing the effect of STIM1, Orai1 and TRCPs related Ca2+ currents in endothelial cells (Moccia: The molecular heterogeneity of store-operated Ca2+ entry in 2 vascular endothelial cells: from Ca2+-selective to non-selective 3 cation channels. Does Orai1 make the difference?). The review paper contains a huge amount of previously published results based on the physiological processes in the scope of the selective or non-selective Ca2+ channels.
The main concerns are:
- In the first view of manuscript it looks crowded and less ordered and thematically sturctured than it is required in the common form of reviews. 23 sides of flowing interpretation, it seems a huge mass of all this topic related published results. Need to reduce it make more informative and well structured.
- In comparison, the comprehensive function of Conclusions is not observable it contains less information than required. It can not give an appropiate answer to the title.
- The Ca2+ influx cause several different processes in the cytoplasm of stressed and recovering cells, and beside the cytoplasmic functions it can modify the whole machinery of nucleus in the collaboration with nuclear and perinuclear Ca2+ sources. Need to interpret all currents in this complex system then focus on the in vitro and in vivo functions of STIM1, Orai1 and TRCPs.
- The biophysical characterization of currents is not identical with couple of electrophysiological values.
- The Figures are not understandable the quality is very week. The message of cartoons is not evident. The meaning of pA/pF values is not interpreted and how can be linked to the cartoon. Need to describe evident and non-evident results which can be found on the figures. Several small mistakes are exsiting (e.g. Protein 4.1 is missing from Fig.3; the sahpe and forms of F-actin is varied on Fig.4).
- Several points of the text are likely random description of references e.g. the function of protein 4.1 and spectrin just appeared randomly then drawn on Fig.4 indepenently of surrounded interpretation in Lines 726-728.
Some minor issues:
- Need to recheck and correct many of typos and minor mistakes as missing space or double point at the end of sentences or occasionally double words.
- In line 163, in the absence of Ca2+ rather than “0Ca2+ ”.
Authors collected huge amount of physiologically relevant results in this manuscript but it seems like a copy paste merged text. However, need to interpret all aspects separately, in vitro and in vivo funtions of ion channels then put them in a well ordered and structured comprehensive review with less randomly collected references, and susequently need to make fascinating clear figures. In this form need a deep major revision and can not be acceptable.
Author Response
Please, see the attachment.

Round 2
Reviewer 2 Report
Dear Authors,
Here is my point-by-point responses.
Dear Reviewer #2,
We are gratefully for your insightful evaluation of our manuscript entitled: “The molecular heterogeneity of store-operated Ca2+ entry in vascular endothelial cells: from Ca2+-selective to non-selective cation channels. Does Orai1 make the difference?” for publication as Review Article in International Journal of Molecular Sciences – Special Issue Age-Related Vascular Physiology.
We respectfully anticipate that we were not able to fully understand some of the comments. Therefore, we tried our best to edit the text according to your indications and carefully addressed all the issues that you kindly raised. In general, we do believe that most of your comments helped us to make the manuscript shorter and more readable. Therefore, we do thank you for this set of observations. Nevertheless, we firmly argue against your comment regarding the presumed copy and paste of other references that (according to your comment) we did throughout the text. We think it is fair to anticipate that we consider this comment as unacceptable and totally out of place.
The review paper contains a huge amount of previously published results based on the physiological processes in the scope of the selective or non-selective Ca2+ channels.
We respectfully, but firmly, disagree with this observation. We do thank the Reviewer for the careful reading of our manuscript. Nevertheless, this is a review article covering the complex issue of the molecular make-up of store-operated Ca2+ entry (SOCE). Therefore, we had to include “a huge amount of previously published results” on this topic and discuss them point-by-point. This is exactly what we meant to do and what we did. We have clarified that the scope of the present review in Section 1, lines 84-86. We respectfully notice that this is the first attempt to shed full light on the evidence that endothelial SOCE is not only mediated by the ICRAC or ISOC, but also by a third membrane conductance with intermediate biophysical properties between them. We kindly invite to directly check this assertion. In a very nice review article (PMID: 28900923) on endothelial SOCE, Lothar Blatter dedicates only 15 lines on page 350 to the existence of ICRAC, ISOC and ICRAC-like currents in vascular endothelial cells (the latter two conductances have not been even mentioned). Another estimated endothelial expert, Klaus Groschner, dedicated to this complex topic only 13 lines on page 476 of his very nice review on the structure and role of SOCE in the cardiovascular physiology (PMID: 28900929). Therefore, our manuscript is the first one to directly address and dedicate so much space to such an important issue to endothelial physiopathology. Our manuscript is the first one to present a side-by-side discussion of the distinct biophysical properties of endothelial SOCE and of the implications of such heterogeneity for the molecular architecture of this vital Ca2+-entry pathway. The discovery by the Trevor Stevens group that, in endothelial cells, Orai1 may associate with TRPC1 and TRPC4 is crucial to understand the structure also of the store-operated current that has long been puzzling endothelial cell physiologists. The Stevens group’s discovery has been largely ignored in the field. For instance, a leading researcher in SOCE structure recently wrote that (PMID: 30521873): “Another prediction was that Orai1 and TRPC1 might form heteromeric channels, with properties distinct from that mediated by Orai1 or TRPC alone [77]. However, neither of these proposals has been further supported by other studies.” Neither of these proposals…. This assertion is not true and simply ignores Stevens’ work on endothelial SOCE. Therefore, ours is not a copy and paste manuscript, as the Reviewer unfairly declares in one of the comments below, but a comprehensive discussion, based upon plenty of published data as a comprehensive review is supposed to do, on the structure and role of endothelial SOCE.
I really apologize but it was a misunderstanding. In this session Authors are on the same opinion as me. There is no contradiction. My state here was a brief summary.
In the first view of manuscript it looks crowded and less ordered and thematically sturctured than it is required in the common form of reviews. 23 sides of flowing interpretation, it seems a huge mass of all this topic related published results. Need to reduce it make more informative and well structured.
We thank the Reviewer also this comment. As outlined above, this review is explicitly dedicated to “bringing order to the distinct mechanisms whereby endothelial SOCE can occur and modulate different functions in the vascular tree from multiple species (e.g., human, mouse, rat, and bovine)” (lines 84-86). Therefore, we explained from the very beginning that this is not a common review article, as, for instance, those nicely written by Blatter and Groschner. However, in order to address the comment raised by the Reviewer, we have curtailed the manuscript by 4 pages although a new Figure, i.e., Figure 5, has been added on request by Reviewer #2. In particular, we have shortened the following sections: Section 1, all Section 2, Section 3.3, and Section 7. We have removed some unnecessary pieces of information, such as the description of the Ca2+ addback protocol (Section 2.2) and the full description of the biophysical properties of TRPC4-containing channels (Section 3.3).
There is a minor improvement with shortening. For focusing on interesing points and make it more readable need to hit the requirements of PRISMA (https://www.mdpi.com/about/article_types).
In comparison, the comprehensive function of Conclusions is not observable it contains less information than required. It can not give an appropiate answer to the title.
We respectfully notice that we could not fully understand this comment. We tried our best to interpret the Reviewer’s thoughts. If we correctly understood the observation, the Reviewer argues that the Conclusions do not provide an appropriate answer to the title. We discussed throughout the manuscript that the physical incorporation of Orai1 (which mediates the Ca2+-selective ICRAC) into a TRPC1/TRPC4-contanining signalplex turns a non-selective cation current (i.e., the ISOC) into a moderately Ca2+-selective one (i.e., the ICRAC-like current). Therefore, in our opinion, the Conclusions provide the answer to the question in the title. To try to address this observation, we changed the title and removed the final question. Of course, we are ready to better discuss with the Reviewer Her/His opinion about our Conclusions. We are sorry that we could not fully interpret Her/His comment at this stage.
The part of Consclusions is very short does not fullfill the requirements. Need to summarize Discussion briefly and provide future directions.
The Ca2+ influx cause several different processes in the cytoplasm of stressed and recovering cells, and beside the cytoplasmic functions it can modify the whole machinery of nucleus in the collaboration with nuclear and perinuclear Ca2+ sources. Need to interpret all currents in this complex system then focus on the in vitro and in vivo functions of STIM1, Orai1 and TRCPs.
We thank the Reviewer for this observation. To the best of our knowledge, and we truly believe we have gone through all the literature covering the role of Orai1, TRPC1 and TRPC4 in vascular endothelial cells, there is no information regarding their role in the modulation of nuclear Ca2+ in the endothelial lineage. Conversely, it has been largely demonstrated that Orai1 and TRPC1, respectively, drive the nuclear translocation of NFAT and NF-kB, which reside in the cytosol under resting conditions and therefore require an increase in intracellular Ca2+ concentration to move into the nucleus. The Qinghua Hu Group nicely showed that the nuclear translocation of NF-kB in endothelial cells is driven by cytosolic, rather than nuclear, Ca2+ oscillations (PMID: 18628303; PMID: 21750195). In the 90s, several studies (PMID: 8178959; PMID: 9501199; PMID: 8891901), including one from our group in 2003 (PMID: 12494452), used confocal microscopy to measure nuclear Ca2+ signals in endothelial cells loaded with Fluo-3 or Fura-2. These oscillations could be supported by SOCE, but these studies were published longer before the discovery of STIM1/Orai1. To address your criticism, we have speculated about the role of endothelial SOCE in the control of nuclear Ca2+ oscillations and functions. In Section 7, we have added the following sub-paragraph: “Fourth, does endothelial SOCE cooperate with perinuclear InsP3Rs via the CICR mechanism to stimulate an increase in perinuclear or nuclear Ca2+? Early studies demonstrated that growth factors induced oscillations in nuclear Ca2+ concentration in both micro- [234] and macrovascular [235] endothelial cells and that these oscillations were abrogated by the pharmacological blockade of SOCE. Future work could assess whether the endothelial SOCE stimulates perinuclear Ca2+ release through InsP3Rs, thereby inducing the nuclear translocation of NFAT and/or NF-kB, and whether the nuclear Ca2+ oscillations sustained by SOCE recruit nuclear localized transcriptional regulators, such as DREAM and myocyte enhancer factor 2C”.
Interestingly, the Qinghua Hu Group published that STIM1 could control nucleoplasmic Ca2+ signalling and gene expression in a RyR-dependent manner (PMID: 26775216). This study was not included in the original submission since it did not directly deal with SOCE, but it has now been discussed in lines 422-429: “In addition, the Hu group reported that, in HUVECs, a Leu-to-Pro substitution in the signal peptide enabled STIM1 to translocate to the nuclear membrane in response to a reduction in [Ca2+]ER [176]. The mutated STIM1 was found to amplify RyRs-mediated increase in nuclear Ca2+ concentration, thereby enhancing the subsequent cAMP responsive element binding protein activity, matrix metalloproteinase-2 (MMP-2) gene expression, and endothelial tube formation. It is noteworthy that the nuclear Ca2+ elevation was independent on SOCE activation [176]. These findings further expand the versatility of STIM1 signalling and place the question as to whether also Orai1 is able to signal in a non-canonical manner in endothelial cells”.
Accepted.
The biophysical characterization of currents is not identical with couple of electrophysiological values.
Honestly, we could not understand this comment. Would the Reviewer be clearer regarding this important point? We do apologise for this misunderstanding.
The biophysical characterization means the application of different type of biophysical methods (e.g. spectroscopies) to study the problem in focus. The description of electrophysiological currents is not biophyisical characterization. Well recommended to use “electrophysiological characterization” or explain all in vitro molecular studies on STIM1 and Orai1.
The Figures are not understandable the quality is very week. The message of cartoons is not evident. The meaning of pA/pF values is not interpreted and how can be linked to the cartoon. Need to describe evident and non-evident results which can be found on the figures. Several small mistakes are exsiting (e.g. Protein 4.1 is missing from Fig.3; the sahpe and forms of F-actin is varied on Fig.4).
We thank the Reviewer for these observations. We do agree with most of them, although we truly believe that the message of the Figures is always straightforward: Figure 1 describes the different store-operated currents that can be activated in endothelial cells; Figure 2 describes the Orai1-dependent mechanism of TRPC1 activation; Figure 3 describes the molecular mechanism of ISOC activation; Figure 4 describes the molecular mechanism responsible for the ICRAC-like currents; and Figure 5 describes the main endothelial functions regulated by each of these current. We truly believe that their message is rather clear.
To address the Reviewer’s criticisms, we amended the Figures in the following manners:
1. The different current density of these three currents depends on the significantly lower single-channel conductance of Orai1 as compared to TRPC1 and TRPC4. This information has been presented in the text and cannot be figuratively depicted. To address this criticism, therefore, we added in the legend of Figure 1 that: “The lower current density of the ICRAC as compared to the ISOC and the ICRAC-like current reflects the single-channel conductance of Orai1 channels, which is 1000 times lower as compared to TRPC channels”.
2. We removed protein 4.1 from Figure 3 since it is not involved in the ISOC signalplex. We do thank the Reviewer for noticing this typo, for which we apologise.
3. We remade Figure 1 to introduce TRPC4, which was actually missing in the original version of the Figure, as well as Figure 3 and Figure 4, in both of which we added some more pieces of information (e.g., the symbols for Ca2+ and the inward currents) and the I-V relationships of, respectively, the ISOC and ICRAC-like current.
4. We have increased to 300 dpi the resolution of each figure.
For these suggestions, we gratefully acknowledge the Reviewer.
As to the shape of F-actin, it is the same in both figures, although it appears smaller in Figure 3.
I would like to keep on my previous opinion that the presentation of Figures need to be improved. Very difficult to understand and find the messages of them based on the small size of components. Makes unconvenient view of impression. However, need to interpret the calculation, meaning and importance of pA/pF. Why can curves be exponential or linear, and why does Isoc go through the origo? F-actin still looks random (e.g. the steps size of helical turns is varied).
Several points of the text are likely random description of references e.g. the function of protein 4.1 and spectrin just appeared randomly then drawn on Fig.4 indepenently of surrounded interpretation in Lines 726-728.
We respectfully disagree with the Reviewer. We firmly argue against the assertion that “some aspects of the text are likely random description of references”. Each reference has been carefully discussed throughout the text. Some of the molecular components of the ISOC signalplex, e.g., protein 4.1 and spectrin, have been inserted exactly where required by the Discussion. They are housekeeping proteins, whose function is perfectly known to all the potential readers of this manuscript (who are supposed to have a strong background in Biology). Since the manuscript is quite long because of the complexity of the topic we addressed, we do not consider it necessary to introduce the functional role of these proteins.
My recommendation is that reduce the size of the text with less number of branching stories, and make it more focused on the entitled topic of “The different roles of Orai1 and TRPC1/TRPC4 in the transition from Ca2+ -selective to non-selective cation channels.”
In line 163, in the absence of Ca2+ rather than “0Ca2+”.
We thank the Reviewer for this observation. However, this part of the manuscript has been removed during the revision process to shorten the manuscript, as recommended by the Reviewer.
Need to recheck and correct many of typos and minor mistakes as missing space or double point at the end of sentences or occasionally double words.
We thank the Reviewer for this observation. The text has been carefully edited.
Authors collected huge amount of physiologically relevant results in this manuscript but it seems like a copy paste merged text.
We firmly argue against this comment, which we judge as totally unfair for the reasons we explained above. We do not write review articles only to publish one more manuscript but to freely express our opinions on the many diverse facets of endothelial Ca2+ signalling. The Reviewer has of course the faculty to reject the present manuscript, but we do not accept and do not tolerate this last comment, which seems to be biased by unmotivated reasons. We understand that the Reviewer has already recommended the manuscript for rejection or is going to do so soon. Nevertheless, scientific dignity and integrity are infinitely more worth than another published manuscript. The corresponding author kindly invites the Reviewer to frankly discuss with him about the reasons of this unfair comment after the revision process.
My recommendation is that reduce the size of the text with less number of branching stories, and make it more focused on the entitled topic of “The different roles of Orai1 and TRPC1/TRPC4 in the transition from Ca2+ -selective to non-selective cation channels.”
Author Response
We are grateful for your insightful evaluation of our manuscript. Please see the attachment for all replies.

Round 3
Reviewer 2 Report
Dear Authors,
Here is my point-by-point responses.
Dear Reviewer #2,
We are grateful for your insightful evaluation of our manuscript entitled: “The molecular
heterogeneity of store-operated Ca2+ entry in vascular endothelial cells: from Ca2+
-selective to
non-selective cation channels. Does Orai1 make the difference?” for publication as Review
Article in International Journal of Molecular Sciences – Special Issue Age-Related Vascular
Physiology.
We addressed most of the criticisms you raised.
There is a minor improvement with shortening. For focusing on interesing points and make
it more readable need to hit the requirements of PRISMA
(https://www.mdpi.com/about/article_types).
We thank the Reviewer for this comment, which gives us the chance to respectfully notice
that the Corresponding Author has published 15 articles on International Journal of Molecular
Sciences, the last one on January 13th
. We know the instruction for the authors of this valuable
journal quite well. According to the link that the Reviewer recalled us (and that we know very
well):
“Reviews offer a comprehensive analysis of the existing literature within a field of study,
identifying current gaps or problems. They should be critical and constructive and provide
recommendations for future research. No new, unpublished data should be presented. The structure
can include an Abstract, Keywords, Introduction, Relevant Sections, Discussion, Conclusions, and
Future Directions, with a suggested minimum word count of 4000 words”.
This is exactly what we did in this manuscript: we provided a comprehensive analysis of the
literature dealing with endothelial SOCE. Moreover, we dedicated a whole paragraph, i.e., Section 7
- The endothelial SOCE machinery: open questions, to highlight the current gaps in the field. We
respectfully notice that the minimum word count amounts to 4000 words. This means that the
journal recommends writing a larger number of words, as we did.
The part of Consclusions is very short does not fullfill the requirements. Need to summarize
Discussion briefly and provide future directions.
We thank the Reviewer for this comment. Actually, we have indicated the future directions
in Section 7 – The endothelial SOCE machinery: open questions, which is quite long. Therefore, we
provided straightforward Conclusions and to summarize the take home message of this manuscript.
To address your criticism, we have expanded the Conclusions to indicate which are, in our opinion,
the most challenging issues in the field.
Accepted.
The biophysical characterization means the application of different type of biophysical
methods (e.g. spectroscopies) to study the problem in focus. The description of electrophysiological
currents is not biophyisical characterization. Well recommended to use “electrophysiological
characterization” or explain all in vitro molecular studies on STIM1 and Orai1.
We have now understood the comment. We thank the Reviewer for having made it clearer.
We replaced “biophysical” with “electrophysiological” throughout the text.
Accepted.
I would like to keep on my previous opinion that the presentation of Figures need to be
improved. Very difficult to understand and find the messages of them based on the small size of
components. Makes unconvenient view of impression. However, need to interpret the calculation,
meaning and importance of pA/pF. Why can curves be exponential or linear, and why does Isoc go
through the origo? F-actin still looks random (e.g. the steps size of helical turns is varied).
We labelled all the Y-axes in the Figures with “current density (pA/pF)” and added a couple
of lines to softly explain its meaning.
Accepted.
As to the IV relationships, we have the feeling that the
Referee is not confident with ion channel electrophysiology. The ICRAC presents an inwardlyrectifying current-to-voltage relationship, which reverses (i.e., changes direction, from inward to
outward) at around +60 mV. This information is described in lines 175-180. TRPC channels in
general present a rather linear IV which reverses at 0 mV (again described in the text, lines 218-
222). Therefore, the IV relationship has to cross the origin. Explaining why CRAC and TRPC
channels present this difference goes far beyond the scope of the present review. Briefly, CRAC
channels, i.e., Orai1, display an unusual pore structure, which does not allow cation efflux from the
cytosol to the extracellular solution. The molecular bases of this behavior have not been understood
yet. TRPC1 and TRPC4 are permeable to both Na+ and K+ almost at the same extent and this is
reflected by their reversal potential, i.e., 0 mV.
As to the current density, we respectfully notice that we have already discussed about the
tiny current density of the endothelial ICRAC in lines 339-357. To address the Reviewer’s concern, in
this Section, we have replaced “amplitude” with “current density”. Furthermore, we have added the
definition of current density in the legend of Figure 1.
Well, if all these interpretations are facts in the Profession of Experts in ion channel electrophysiology, I can agree. If it is more related to the special properties of these type of ion-channels, I guess Authors need to describe them in the text for a better understand.
F-actin does not look random either in Figure 3 or in Figure 4. In Figure 4, the involvement
of protein 4.1 has been described. This protein does not seem to be involved in the ISOC and,
therefore, has not been added in Figure 3. The signalplex is Figure 3 is very rich and involves
caveolin-1 and InsP3Rs. This is why the size of each protein component is smaller as compared to
Figure 4A, in which the ordered F-actin cytoskeleton beneath the plasma membrane has been put in
evidence.
At Fig.4 „F-actin still looks random (e.g. the steps size of helical turns is varied).”
Need to improve the Figures. There is no reason why are these compressed in small area and all labeles are loitering around structures. TRCP1 size varies with vesicles and PM localization. Need to improve them to less crowded and more focused on the molecular events, emphasize the meaning of the arrows. The image of ER is less important than the arrows within the Ca2+ fluxes. Arrows need to start from the ion not from the open field. At Fig.3 we cannot see any event, this is the most crowded figure. At Fig. 2 purple closed circles are not labeled.
Round 4
Reviewer 2 Report
Dear Authors,
I am sure that your manuscript shows a great improvement.
Congratulation! All the Bests!